# Nitric oxide inhibits ten-eleven translocation DNA demethylases to regulate 5mC and 5hmC across the genome

Marianne B. Palczewski [1], Hannah Petraitis Kuschman[1], Brian M. Hoffman [2], Venkatesan Kathiresan [2], Hao Yang[2], Sharon A. Glynn[3], David L. Wilson[4], Eric T. Kool [4], William R. Montfort [5], Jenny Chang[6], Aydolun Petenkaya[7], Constantinos Chronis[8], Thomas R. Cundari[9], Sushma Sappa[10], Kabirul Islam[10], Daniel W. McVicar [11], Yu Fan [12], Qingrong Chen[12], Daoud Meerzaman[11], Michael Sierk [11] & Douglas D. Thomas [1] ✉

DNA methylation at cytosine bases (5-methylcytosine, 5mC) is a heritable epigenetic mark regulating gene expression. While enzymes that metabolize 5mC are well-characterized, endogenous signaling molecules that regulate DNA methylation machinery have not been described. We report that physiological nitric oxide (NO) concentrations reversibly inhibit the DNA demethylases TET and ALKBH2 by binding to the mononuclear non-heme iron atom forming a dinitrosyliron complex (DNIC) and preventing cosubstrates from binding. In cancer cells treated with exogenous NO, or endogenously synthesizing NO, 5mC and 5-hydroxymethylcytosine (5hmC) increase, with no changes in DNA methyltransferase activity. 5mC is also significantly increased in NO-producing patient-derived xenograft tumors from mice. Genome-wide methylome analysis of cells chronically treated with NO (10 days) shows enrichment of 5mC and 5hmC at gene-regulatory loci, correlating with altered expression of NO-regulated tumor-associated genes. Regulation of DNA methylation is distinctly different from canonical NO signaling and represents a unique epigenetic role for NO.

DNA methylation at the 5-carbon position of cytosine bases (5mC) is propagated through cell division and is considered a key epigenetic mechanism for cellular memory of transcriptional states[1–3]. 5mC at promoter CpG islands is associated with repression of gene expression, X-chromosome inactivation, and genomic imprinting. Recent research has demonstrated a more complex relationship between DNA methylation and gene expression that differs depending on the genomic context in which it occurs and is not restricted to

[1]Department of Pharmaceutical Sciences, University of Illinois Chicago, College of Pharmacy, Chicago, IL, USA. [2]Department of Chemistry, Weinberg College of Arts and Sciences, Northwestern University, Evanston, IL, USA. [3]Discipline of Pathology, University of Galway, College of Medicine, Nursing and Health Sciences, School of Medicine, Galway, Ireland. [4]Department of Chemistry, Stanford University, School of Humanities and Sciences, Stanford, CA, USA. [5]Department of Chemistry and Biochemistry, University of Arizona, Tucson, AZ, USA. [6]Dr. Mary and Neal Cancer Center at Houston Methodist, Weill Cornell Medical College, Houston, NY, USA. [7]Department of Biomedical Engineering, University of Illinois Chicago, College of Engineering, Chicago, IL, USA. [8]Department of Biochemistry and Molecular Genetics, University of Illinois Chicago, College of Medicine, Chicago, IL, USA. [9]Department of Chemistry, University of North Texas, Denton, TX, USA. [10]Department of Chemistry, University of Pittsburgh, Pittsburgh, PA, USA. [11]Cancer Innovation Laboratory, National Cancer Institute, Center for Cancer Research, Frederick, MD, USA. [12]National Cancer Institute, Center for Biomedical Informatics and Information Technology, Bethesda, USA. ✉e-mail: ddthomas@uic.edu

transcriptional regulation and silencing at promoter regions[4,5]. For example, 5mC methylation at enhancer regions can repress enhancer activity whereas gene body methylation can facilitate transcriptional elongation and is associated with actively transcribed genes[6].

Regardless of the functional consequences of 5mC on gene expression, specific DNA methylation states are established and maintained in cells by the concerted activities of methyl-modifying enzymes. This enzymatic machinery includes DNA methyltransferases (DNMTs), which install 5mC on DNA, and Ten-Eleven Translocation (TET) enzymes that facilitate active demethylation. There are three TET enzyme paralogs (TET1, TET2, and TET3) and each can sequentially oxidize 5mC to 5-hydroxymethylcytosine (5hmC), 5-formylcytosine (5fC) and 5-carboxylcytosine (5caC). This is followed by base excision repair by thymine DNA glycosylases (TDGs), which results in an unmethylated cytosine[7]. Emerging studies indicate that these more oxidized forms of 5mC (i.e., 5hmC, 5fC, and 5caC) can similarly participate in transcriptional activation or repression when located at promoter regions, within gene bodies, or other non-promoter genomic regions[8,9].

Besides its crucial role in gene regulation, DNA methylation can also occur by pathological alkylation damage from exposure to environmental or therapeutic alkylating agents, or from endogenous metabolic processes. This type of alkylation DNA damage methylation occurs on any of the 4 DNA bases but is most prevalent on adenine (as 1-methyladenine (1mA)) and on cytosine (as 3-methylcytosine (3mC)). To safeguard genomic integrity, repair pathways have evolved to recognize these cytotoxic alkylation-induced DNA lesions. One key enzyme in these processes is human AlkB homolog 2 (ALKBH2), a DNA repair enzyme that recognizes and demethylates 1mA and 3mC back to adenine and cytosine, respectively[10]. Both alkylation-mediated DNA methylation and gene-regulatory DNA methylation are associated with numerous disease states including cancers. In cancer, changes in 5mC DNA methylation can deactivate tumor suppressor genes or activate oncogenes, while 1mA and 3mC are mutagenic[11].

Despite the importance of methylated DNA bases and their oxidized derivatives in controlling normal gene expression or facilitating oncogenic transcriptional states, there is a lack of mechanistic knowledge regarding endogenous molecular regulators of DNA methyl-modifying machinery. One such potential regulator, which has not been extensively studied in the context of epigenetics, is nitric oxide (NO), an endogenously produced free radical signaling molecule known for its diverse physiological roles, including neurotransmission, immune responses, and vascular homeostasis[12]. NO is a reactive molecule and has a short biological half-life[13], yet under cellular conditions, its unique chemistry restricts its reactions to only metals and other free radicals, including oxygen ($O_2$)[14]. To mediate its signaling effects, NO primarily targets metalloproteins, particularly those coordinating iron at their active sites, regulating their functions through the formation of complexes such as iron nitrosyl (Fe(II) = NO) in heme proteins, or dinitrosyl iron complexes (DNIC) in non-heme proteins[15]. Alternatively, NO can indirectly regulate enzyme function post-translationally through complex reactions that form adducts containing nitrogen oxide functional groups, such as S-nitrosothiols (RSNO)[12,16,17]. Regardless of NO's mechanism of action, the specificity of NO signaling, which can either activate or inhibit protein activity, is governed by various microenvironmental factors, including the type and abundance of target proteins, the concentration and duration of NO synthesis, the redox state of the cell, and local oxygen concentrations. These parameters influence the chemical reactions of NO which result in signaling that is both highly regulated and context-specific.

In addition to the critical roles NO plays in maintaining normal physiological homeostasis, dysregulated NO synthesis and signaling are implicated in various pathological conditions, particularly cancers. Elevated NO production, often driven by upregulation of the inducible form of nitric oxide synthase (NOS2), is associated with altered tumor gene expression, poor patient outcomes, increased mortality, and resistance to chemotherapy across various cancer types, including triple-negative breast cancer (TNBC)[18–28], lung[29–31], prostate[32,33], brain[34], colon[35,36], melanoma[37–39], and liver[40,41]. However, despite major advances in our understanding of NO signaling, it remains insufficient to fully explain the complexity of NO's diverse effects in disease.

Dysregulated NO production and perturbations in DNA methylation patterns are both associated with many of the same pathologies, yet a direct causal mechanism linking NO synthesis to altered DNA methylation patterns has not been described. Our previous research demonstrated a unique role for NO as an endogenous epigenetic regulator of gene expression by controlling histone post-translational modifications[42,43] and mRNA methylation (m6A) patterns[44]. We demonstrated mechanistically that physiologic NO concentrations could inhibit the catalytic activities of histone lysine demethylases (KDM) and the RNA demethylase fat mass and obesity associated protein (FTO)[45–47] by forming DNIC at their active sites. Since KDM and FTO demethylases belong to the same family of Fe(II)/2-oxoglutarate (OG)-dependent oxygenases as the DNA demethylases TET and ALKBH2, we hypothesize that NO can similarly inhibit TET and ALKBH2 to change DNA methylation patterns.

This study investigates NO as an endogenous regulator of DNA methylation, with a focus on triple-negative breast cancer (TNBC) models. This focus is driven by the observation that patients with this molecular subtype that harbor NOS2-expressing tumors have a significantly higher mortality rate than patients with tumors that do not express NOS2[27]. Although the ability to correctly assign the transcriptional regulation of a particular gene to the local (or distant) presence of DNA methylation has remained surprisingly limited[48], our data suggest that deleterious phenotypic effects of NO are partially determined by transcriptional reprogramming mediated by TET inhibition and the resultant changes in the DNA methylation landscape.

## Results

### Determination of NO as an inhibitor of TET and ALKBH2 demethylase activity in vitro

Our research has shown that NO can inhibit the catalytic activities of other epigenetic Fe(II)/2-OG-dependent enzymes: KDM3A, a histone demethylase[42,43], and FTO and ALKBH5, mRNA demethylases[44]. In humans, active DNA demethylation is catalyzed by 2 members of the Fe(II)/2-OG-dependent family of oxygenases: TET (paralogs TET1, TET2 & TET3) and ALKBH2. To test whether NO could inhibit TET catalytic activity we incubated the purified catalytic domain of human TET2 enzyme with all cofactors (2-OG, Fe(II), ascorbate), substrate (synthetic 5mC-DNA oligo), and NO (the NO donor Sper/NO, 50–500 µM). After a 3 h incubation, we measured relative amounts of 5mC, the substrate for TET2, and its initial oxidation product 5hmC at each NO (Sper/NO) concentration using a MALDI-TOF-MS-based assay[49] (Fig. 1a). Using a broader range of physiologic/pathologic NO concentrations, we determined that NO inhibited TET2 demethylase activity with a half-maximal inhibitory concentration ($IC_{50}$) of 165 µM for Sper/NO (Fig. 1b).

Because of the small reaction volumes used for these experiments, we were not able to directly measure steady-state NO concentrations ($[NO]_{ss}$). NO donor compounds, such as Sper/NO, are very useful for studying biological effects of NO because of their predictable release kinetics. However, the steady-state NO concentration cannot be accurately extrapolated from the starting concentration of the NO donor. To estimate the inhibitory $[NO]_{ss}$ for TET, we solved a system of differential equations that collectively considered dynamic changes in the concentrations of NO donors, NO, and $O_2$ over time. In an aqueous solution with isolate enzyme, the $[NO]_{ss}$ is determined by the difference in the rate of NO liberation from the donor and the rate of NO consumption (predominantly auto-oxidation). Based on these

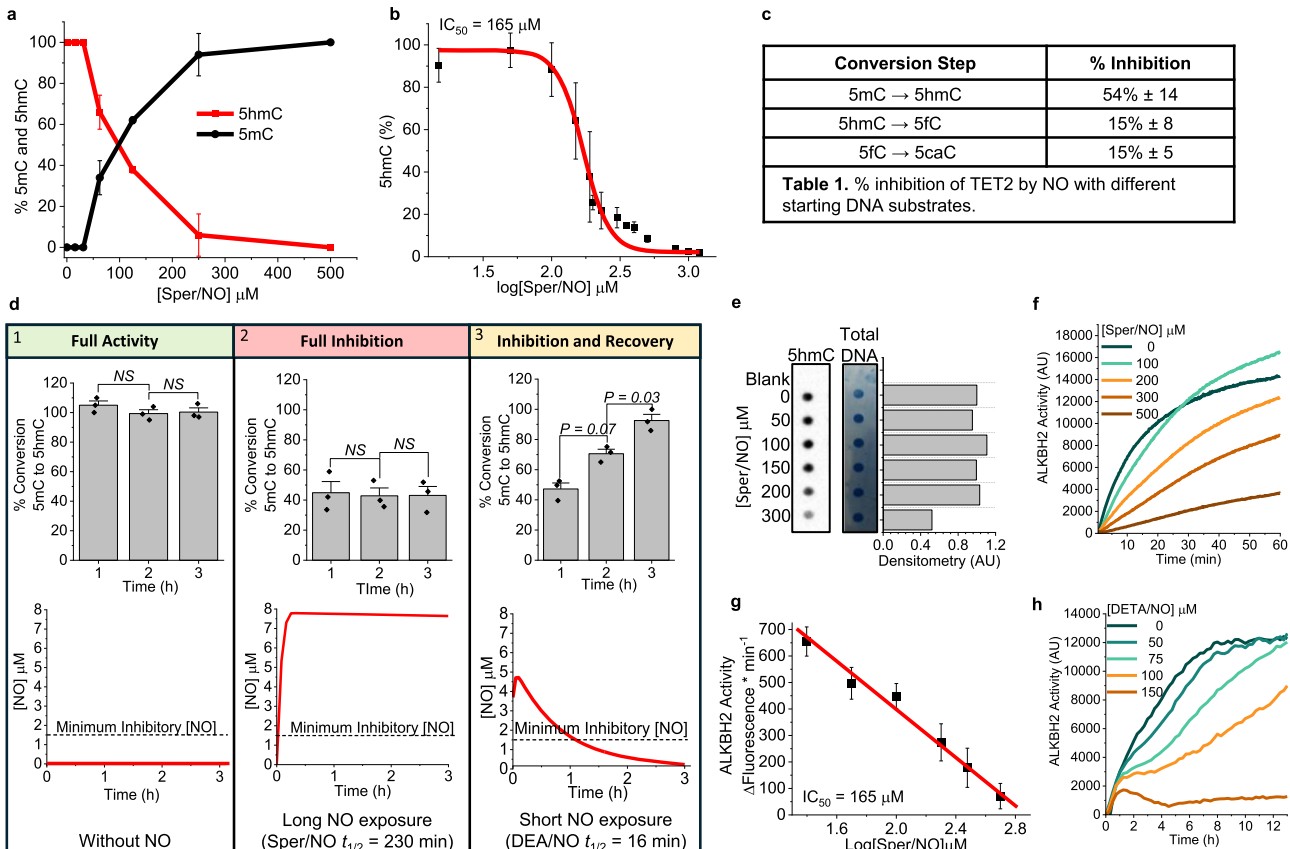

| Conversion Step | % Inhibition |
|---|---|
| 5mC → 5hmC | 54% ± 14 |
| 5hmC → 5fC | 15% ± 8 |
| 5fC → 5caC | 15% ± 5 |

**Table 1.** % inhibition of TET2 by NO with different starting DNA substrates.

**Fig. 1 | NO reversibly inhibits Fe(II)/2-OG-dependent DNA demethylases.** TET2 demethylase activity (**a**–**e**): **a** Conversion of 5mC (black line) to 5hmC (red line) when human TET2 was incubated with NO (Sper/NO; 0–500 μM; 3 h). Demethylase activity assay was initiated with addition of a 5mC-double-stranded DNA substrate, n = 3. **b** TET2 activity (5hmC formation) measured over a range of Sper/NO concentrations (0–1.5 mM), $IC_{50}$ = 165 μM Sper/NO, n = 2. **c** % inhibition of TET2: TET2 was incubated with 165 μM Sper/NO for 3 h, the reaction was started by the addition of one of three different DNA substrates for TET2 (DNA oligos containing 5mC, 5hmC, or 5fC), n = 2. **d** TET2 demethylase activity (5hmC product formation, ELISA, n = 3) was measured at 3 time points (1, 2, & 3 h) and under three different conditions: Panel 1 TET2 incubated without NO: Full Activity, Panel 2 TET2 incubated continuously with NO (300 μM Sper/NO): Full Inhibition, and Panel 3: TET2 incubated with NO for a short duration (25 μM DEA/NO): Inhibition and Recovery. The top bar graphs are the % product formation at each time point. The bottom graphs

are the simulated $[NO]_{ss}$ (red line) under each reaction condition, data (**a**–**d**) are mean ± SEM *P*-values determined by unpaired (**a**–**c**) or paired (**d**) two-tailed Student's t-tests. **e** TET2 activity in the presence of freshly isolated genomic DNA (1 μg) and NO (Sper/NO 0–300 μM, 3 h), 5hmC measured by dot-blot hybridization using anti-5hmC antibody and total DNA measured by methylene blue, representative blot and densitometry shown, n = 3. **f** ALKBH2 activity: recombinant ALKBH2 was incubated with NO (Sper/NO 0–500 μM) and the fluorescent probe. ALKBH2 activity (demethylation of the fluorescent probe) was monitored for 1 h by fluorescence spectroscopy at 480 nm. **g** $IC_{50}$ determination from data in panel **f**, mean ± S.D., **h** Cellular extracts containing ALKBH2 were incubated with the fluorescent probe and the slow NO-releasing donor DETA/NO ($t_{1/2}$ = 22 h, 0–150 μM), demethylation was measured for 12 h (**f**, **h**: representative graphs, n = 3 separate experiments). NS not significant, AU arbitrary units. Source data are provided as a Source Data file.

simulations, the $IC_{50}$ for $[NO]_{ss}$ was approximately 3 μM for isolated TET2 enzyme under our assay conditions (Supplementary Fig. 1a, Supplementary Data 1). Having established that NO could inhibit the first step of TET's three sequential oxidations (5mC ⇒ 5hmC) we next tested whether NO could also inhibit the stepwise oxidation of 5hmC ⇒ 5fC, and in a separate reaction, the conversion of 5fC ⇒ 5caC. We conducted two separate reactions, one that used 5hmC as the starting TET2 substrate and the other that used 5fC as the initial substrate. Both reactions were treated with the $IC_{50}$ concentration of Sper/NO for TET2 (165 μM). When the reaction was started with either 5hmC-DNA or 5fC-DNA as the TET2 substrate, the enzymatic conversion to 5fC or 5caC was inhibited by 15% (Fig. 1c).

We next asked whether inhibition of TET2 by NO was permanent or reversible. To address this question, we measured TET2 activity (conversion of 5mC to 5hmC) under different conditions of NO concentration and duration of exposure. The concentrations of NO and the durations of NO exposure were established using two different NO donor compounds: DEA/NO, a short-acting NO donor ($t_{1/2}$ ≈ 16 min at 25 °C), and Sper/NO, a longer-acting NO donor ($t_{1/2}$ ≈ 230 min at 25 °C)[50]. Mathematical modeling was used to

simulate theoretical steady-state NO concentrations from each NO donor under each experimental reaction condition (bottom graph panels in Fig. 1d). The simulated kinetics and steady-state NO concentrations were similar to what we previously determined experimentally with these NO donors[15,50–52]. Predicted changes in the concentrations of $O_2$ and NO donor compounds were also modeled (Supplementary Fig. 1b, c and Supplementary Data 1). When we measured TET2 demethylase activity in the absence of NO, TET2 was fully active and resulted in 100% conversion of 5mC ⇒ 5hmC within the first hour (Full Activity, Fig. 1d. **1**). We then exposed TET2 to a continuous $[NO]_{ss}$ for the entire 3 h duration of the experiment. Initially, TET2 exhibited demethylase activity (~40% product formation) until the 1 h point when an inhibitory concentration of NO had accumulated within the reaction vessel. At 1 hour and beyond TET2 activity was 100% suppressed and no further 5mC demethylation was measured (Full Inhibition, Fig. 1d. **2**). Next, we incubated TET2 with the NO donor DEA/NO, which, due to its rapid release kinetics, produced an initial burst of NO that disappeared by the end of the 3 h experiment. Under this condition, TET2 demethylase activity was inhibited at 1 h when the NO concentration was high. However, by

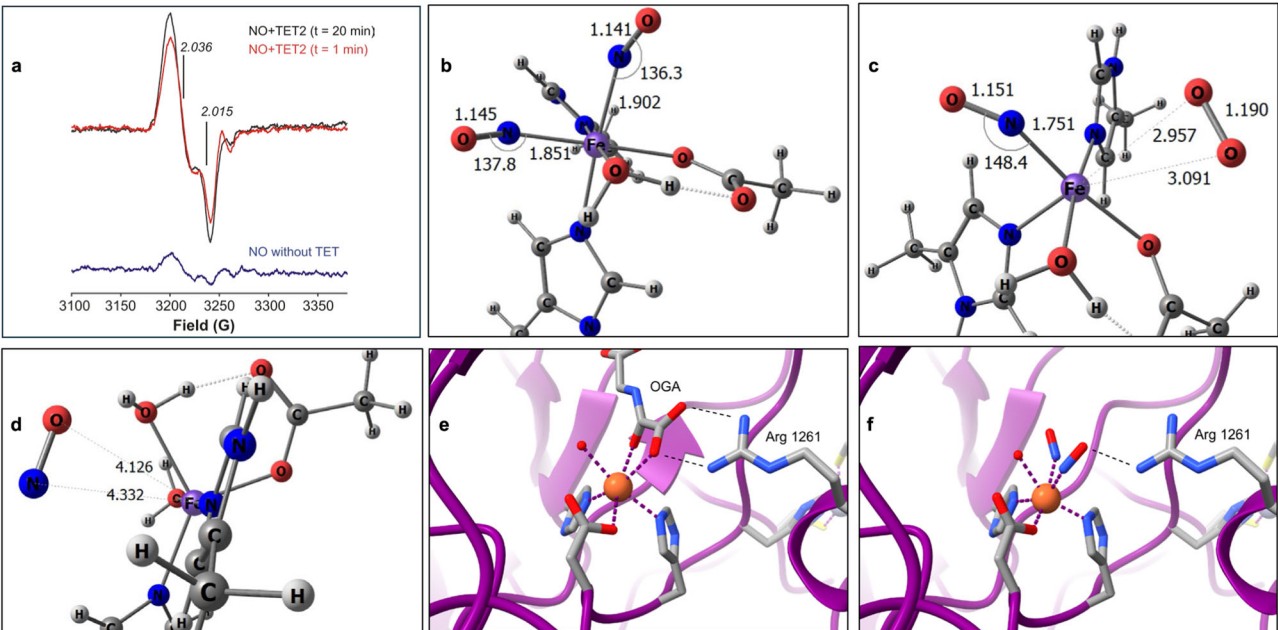

**Fig. 2 | NO forms a DNIC at the mononuclear non-heme iron atom of TET2.**
**a** Representative X-Band (77 K) EPR spectra of truncated TET2 protein incubated with NO (Sper/NO, 100 μM) and all substrates and cofactors for 1 min (red line) or 20 min (black line). (Blue line) is the complete reaction without TET2 at 20 min, n = 3. Spectrum is indicative of a non-Heme NO-bound DNIC. DFT computations (**b–d**): **b** Core of the ωB97xD/def2-svp optimized geometry of triplet dinitrosyl complex $[Fe^{II}(OAc)(Im)_2(NO)_2(OH_2)]^+$, OAc = acetate, Im = N-methyl-imidazole. Bond lengths and angles in Å and °, respectively. **c** Core of the ωB97xD/def2-svp optimized geometry of $[Fe(OAc)(Im)_2(NO)(O_2)(OH_2)]^+$ and (**d**) $[Fe(OAc)(Im)_2(NO)(OH_2)_2]^+$. **e** Crystal structure of TET2 in complex with N-oxalylglycine (OGA), a 2-OG analog (PDB file 4nm6). OGA forms two critical hydrogen bonds with a conserved arginine (Arg 1261 in TET2). **f** Two NO molecules replace the two oxygens from OGA that coordinate with Fe(II). Shown also are Arg 1261, which may stabilize NO binding, a water molecule in the sixth coordination site, and coordinating residues His 1382, Asp 1384, and His 1881.

2 h, when the NO concentration had decreased, we observed recovery of enzyme activity, and at 3 h the reaction was 100% complete (Inhibition and Recovery, Fig. 1d. **3**). Since TET2 enzymatic activity was restored between 1 and 2 h, this suggested that the minimum inhibitory concentration for TET2 was within the range of 0.6–1.65 μM NO. The observed recovery of enzyme activity following NO disappearance suggested that the inhibitory effect is reversible and not due to permanent covalent modifications or degradation of the enzyme. Lastly, instead of using a synthetic 5mC-DNA oligo substrate, we tested NO-mediated TET2 inhibition using its biological substrate: genomic DNA (Fig. 1e). NO inhibited TET2-catalyzed conversion of 5mC to 5hmC in genomic DNA at similar Sper/NO concentrations as when 5mC-DNA oligos were used as substrates.

Having established that NO could inhibit TET2, we tested whether NO could similarly inhibit ALKBH2, another Fe(II)/2-OG-dependent DNA demethylase. ALKBH2 activity was measured in real-time using a fluorescence assay[53]. This assay used a fluorogenic 1-methyladenine probe that has a > 10-fold increase in fluorescent signal intensity when the alkyl group is demethylated by ALKBH2. We incubated recombinant ALKBH2 and all cofactors (2-OG, Fe(II), ascorbate) with the methylated probe substrate and NO (Sper/NO, 100–500 μM) for 60 min (Fig. 1f). Under these conditions, NO inhibited ALKBH2 demethylase activity in a concentration-dependent manner with an $IC_{50}$ for Sper/NO of 165 μM (Fig. 1g). This $IC_{50}$ for Sper/NO was the same as what we measured for TET2, suggesting that the $IC_{50}$ for NO is also similar for the two proteins (~3 μM $[NO]_{ss}$). Next, we isolated nuclear and cytosolic extracts containing endogenous ALKBH2 from cultured MDA-MB-231 breast cancer cells to test whether NO could similarly inhibit ALKBH2 derived from biological sources. The combined extracts were exposed to low steady-state concentrations of NO using the NO donor DETA/NO ($t_{1/2} \approx$ 22 h, 50–150 μM) and demethylase activity was monitored for 12 h using the fluorescent probe method

(Fig. 1h). Under these conditions, ALKBH2 demethylase activity was inhibited in a dose-dependent manner. We estimated that the steady-state concentrations of NO over this range of DETA/NO concentrations to be within physiological levels ($[NO]_{ss}$ ~ 600 nM, Supplementary Fig. 1d, Supplementary Data 1, and as we have previously measured[15,52]). Collectively, these results indicate that NO is a potent inhibitor of Fe(II)/2-OG-dependent DNA demethylases TET2 and ALKBH2. For subsequent studies, we solely focused on TET enzymes because of their gene regulatory functions (rather than ALKBH2 which is part of the DNA damage response).

## Nitric oxide forms a dinitrosyl iron complex at the mononuclear non-heme iron atom in TET2

A critical step in the TET-catalyzed DNA demethylation reaction is binding $O_2$ to the non-heme iron atom[54]. We hypothesized that because of NO's structural and bonding similarity to $O_2$ that NO would compete with $O_2$, inhibiting TET by forming the more stable mononitrosyl complex at the iron site. To test whether NO interacts with the iron center, we conducted electron paramagnetic resonance (EPR) studies of TET2 treated with NO in the presence of all cofactors and substrate. We ran two identical reactions in parallel and stopped them at different time points (< 1 and 20 minutes). EPR spectra at both time points indicated that the enzyme does not form the expected S = 3/2 mononitrosyl complex (Fe(II)-NO), as shown by the absence of its intense $g_\perp$ ~ 4.1 signal (Supplementary Fig. 2a), and instead revealed the characteristic S = ½ signal of a non-heme dinitrosyl iron complex (DNIC, $[Fe(NO)_2]$9), with $g_\perp$~2.03, $g_\parallel$ ~ 2.01 (Fig. 2a)[15,55,56]. Kinetically, the reaction was ~80% complete by the time all reactants were added, as shown by the small increase in the EPR intensity between samples frozen at 1 min and 20 min. The signature EPR spectrum of DNIC was almost undetectable when the TET2 enzyme was omitted from the complete reaction mixture indicating that the observed DNIC signal is

associated with the enzyme and not with a complex formed in solution (Fig. 2a). Moreover when the NO source was depleted and NO disappeared from the solution the TET2-DNIC EPR signal also disappeared (Supplementary Fig. 2b), which coincided with gain of TET2 demethylase activity (Fig. 1d. **3**).

## Density functional computations support the formation of a DNIC at the catalytic iron atom in TET2

Because the DNIC was formed instead of the mononitrosyl, a series of density functional theory (DFT) computations were performed to investigate the relative affinity for binding of NO versus $O_2$ and water in the TET2 resting state. An active site model of TET2 was generated akin to that reported by Lu et al.[57]. The charge of the model DNIC complex was adjusted to yield a neutral {Fe(NO)$_2$}9 electron count, i.e., d6 Fe(II) + 2 NO $\pi^*$ $e^-$ + 1 additional $e^-$. Additional simulations were performed to assess NO binding to models in which 2-OG was ligated to the inner coordination sphere, but these were largely inconclusive apart from indicating that NO is bound more weakly to Fe(II) after ligation of 2-OG. The iron in the enzyme is coordinated by two histidine (His) and one aspartate (Asp) residue. To mimic the pertinent amino acid side chains, Asp was modeled by an acetate (OAc$^-$) and His was modeled by N-methyl-imidizaole (Im). It was assumed that the Asp and His model ligands would maintain a *fac* configuration.

The neutral {Fe(NO)$_2$}9 dinitrosyl complex Fe(OAc) (Im)$_2$(NO)$_2$(OH$_2$) was chosen for modeling studies; the geometry is shown (Fig. 2b). Among all the spin states assessed for Fe(OAc) (Im)$_2$(NO)$_2$(OH$_2$), the triplet (intermediate spin, 2 unpaired electrons) is predicted to be the lowest in free energy relative to the high spin (quintet) and low spin (singlet) states[56]. DNIC, with an extra electron in the complex, is a typical configuration and has previously been described as having a triplet state[55]. While the source of this extra electron is not yet known, many reductants are available in biological systems. The lowest energy coordination isomer had one NO trans to OAc and the other trans to Im (Fig. 2b). The OAc that mimics the Asp side chain forms a strong hydrogen bond with the ligated water. Given the experimental spectroscopic observations, it was investigated whether one of the NO ligands of Fe(OAc)(Im)$_2$(NO)$_2$(OH$_2$) could be displaced by either water or dioxygen; thus, computed DFT ground states of Fe(OAc)(Im)$_2$(NO)(O$_2$)(OH$_2$) and Fe(OAc)(Im)$_2$(NO)(OH$_2$)$_2$ were sought; the lowest energy geometries are shown in Fig. 2c and d, respectively. In both cases, geometry optimizations initiated from six-coordinate, pseudo-octahedral starting guesses yielded minima with a weakly bound exogenous ligand. For Fe(OAc)(Im)$_2$(NO)(OH$_2$)$_2$, the NO ligand is ejected from the inner coordination sphere upon geometry optimization, Fe$\cdots$NO = 4.13 Å (Fig. 2d). For Fe(OAc)(Im)$_2$(NO)(O$_2$)(OH$_2$) (Fig. 2c), the complex barely maintains an octahedral geometry with the dioxygen very weakly bonded (Fe$\cdots$O$_2$-2.95 Å). The tenuous nature of $O_2$ binding in Fe(OAc)(Im)$_2$(NO)(O$_2$)(OH$_2$) is further indicated by the near complete lack of any spin delocalization from the $O_2$ to the complex ($r_{spin}$(O$_2$) = 1.99 $e^-$) and the computed free energy for NO/H$_2$O exchange, $\Delta G$ = +23.0 kcal/mol, indicating that NO binds much more tightly than water. The NO/$O_2$ exchange free energy is essentially thermoneutral, $\Delta G$ = −0.3 kcal/mol. In conjunction with the weak $O_2$ binding indicated by the long Fe$\cdots$O$_2$ bond length of the optimized geometry of Fe(OAc)(Im)$_2$(NO)(O$_2$)(OH$_2$) (Fig. 2c), the DFT results suggest that $O_2$ does not readily displace NO, perhaps except at high $O_2$ partial pressures, and rationalizes the presence of only the dinitrosyl complex as seen in the EPR spectra (Fig. 2a).

Based on the DFT results, further modeling was done for the DNIC core and the crystal structure of TET2 in complex with N-oxalylglycine (OGA), a 2-oxoglutarate analog[58] (PDB ID 4NM6, Fig. 2e). The OGA complex has iron coordinated to two histidine residues and an aspartate, a typical arrangement for Fe(II)/2-OG-dependent DNA demethylases. OGA is coordinated to the iron through two oxygens and stabilized through hydrogen bonding to Arg 1261 and a second

arginine. A water molecule occupies the sixth coordination site in the structure, which is replaced by dioxygen during catalysis. In the DNIC model, the NO molecules replace OGA and occupy similar positions to the OGA-coordinating oxygens. One of these is close to Arg 1261, which may hydrogen bond to the NO and stabilize overall negative charge buildup on the NO ligands in the DNIC (Fig. 2f).

## Nitric oxide increases 5mC in DNA from human cancer cells

Having established that NO was a direct and potent inhibitor of TET enzymes under isolated conditions, the next step was to investigate whether NO could inhibit endogenous cellular TET enzymes and to determine if this would regulate nuclear DNA methylation. We selected four human cancer cell lines derived from aggressive tumor types that are known to express NOS2 and synthesize NO in vivo (MDA-MB-231 and MDA-MB-668 are TNBC, PC-3 is prostate, and U251 is glioblastoma). Because these cells do not synthesize NO in culture, we exposed them to nM concentrations of NO using the NO donor DETA/NO (100 μM; 24 h) and measured 5mC-DNA (Fig. 3a). In all cell lines NO significantly increased global 5mC in DNA. Among NO-associated cancers, TNBC patients who harbor NOS2-expressing tumors have significantly worse prognoses. For this reason, we conducted all subsequent experiments using models of TNBC.

Under biological conditions, DNA methylation patterns are faithfully maintained over multiple cell generations as a form of epigenetic inheritance. Therefore, we developed a cell model to study DNA methylation responses to NO after multiple cell generations; this more accurately mimics the microenvironment of NOS2-expressing tumors in vivo where cells are exposed to chronic NO synthesis. We treated two TNBC cell lines with low physiologic steady-state concentrations of NO for 10 days (~12 cell doublings (NO did not alter the doubling rate)) and examined long-term heritable DNA methylation patterns. In the NO-treated cells, there was a significant increase in 5mC in DNA (Fig. 3b), as well as an increase in 5hmC in DNA, the first oxidation product of TET (Fig. 3c). Cells not treated with NO had no change in 5mC/5hmC after 10 days, suggesting 5mC increases were not attributable to epigenetic drift. To mimic the endogenous NO production observed in tumors, we transfected MDA-MB-231 cells with a human *NOS2* gene (or empty vector control (VC)) (Fig. 3d). Nitrite, a specific indicator of accumulative NO synthesis, was quantified in the media of both cell lines after 24 and 48 h. NO was detectable solely in the *NOS2*-transfected cells, not in those transfected with the empty vector plasmid or in *NOS2*-transfected cells treated with a pan-NOS inhibitor (L-NMMA) (Fig. 3e). When 5mC-DNA levels were measured at 24 and 48 h in the same cell lines, 5mC increased only in the NO-producing cells but not in control cells or those treated with L-NMMA (Fig. 3f).

5mC is catalytically installed on DNA by DNA methyltransferase enzymes (DNMT) which, along with TET enzymes, maintain steady-state 5mC levels. NO-dependent increases in 5mC could therefore be due to increased DNA methyltransferase activity rather than inhibition of TET DNA demethylase activity. To test this, we treated MDA-MB-231 cells with NO and one of two compounds that decrease DNMT activity: either a DNA methyltransferase 1 (DNMT1) inhibitor (5-Azacytidine (AZA)) or a competitive inhibitor of methionine adenosyltransferase (MAT) (cycloleucine (CL)). CL depletes the cell of S-adenosylmethionine (SAM), the substrate for DNA methyltransferases (Fig. 3g). In cells treated with either AZA or CL alone, a significant reduction in global 5mC levels was observed as expected[59], but when NO was present during either CL or AZA treatment, 5mC levels remained elevated (Fig. 3h). Another mechanism by which NO could increase 5mC is by changing the expression levels of DNA methyl-modifying enzymes (i.e., increasing DNMT or decreasing TET expression). To examine this possibility, we treated TNBC cell lines with NO for 10 days and measured the protein expression levels of DNA methyltransferases (DNMT1,

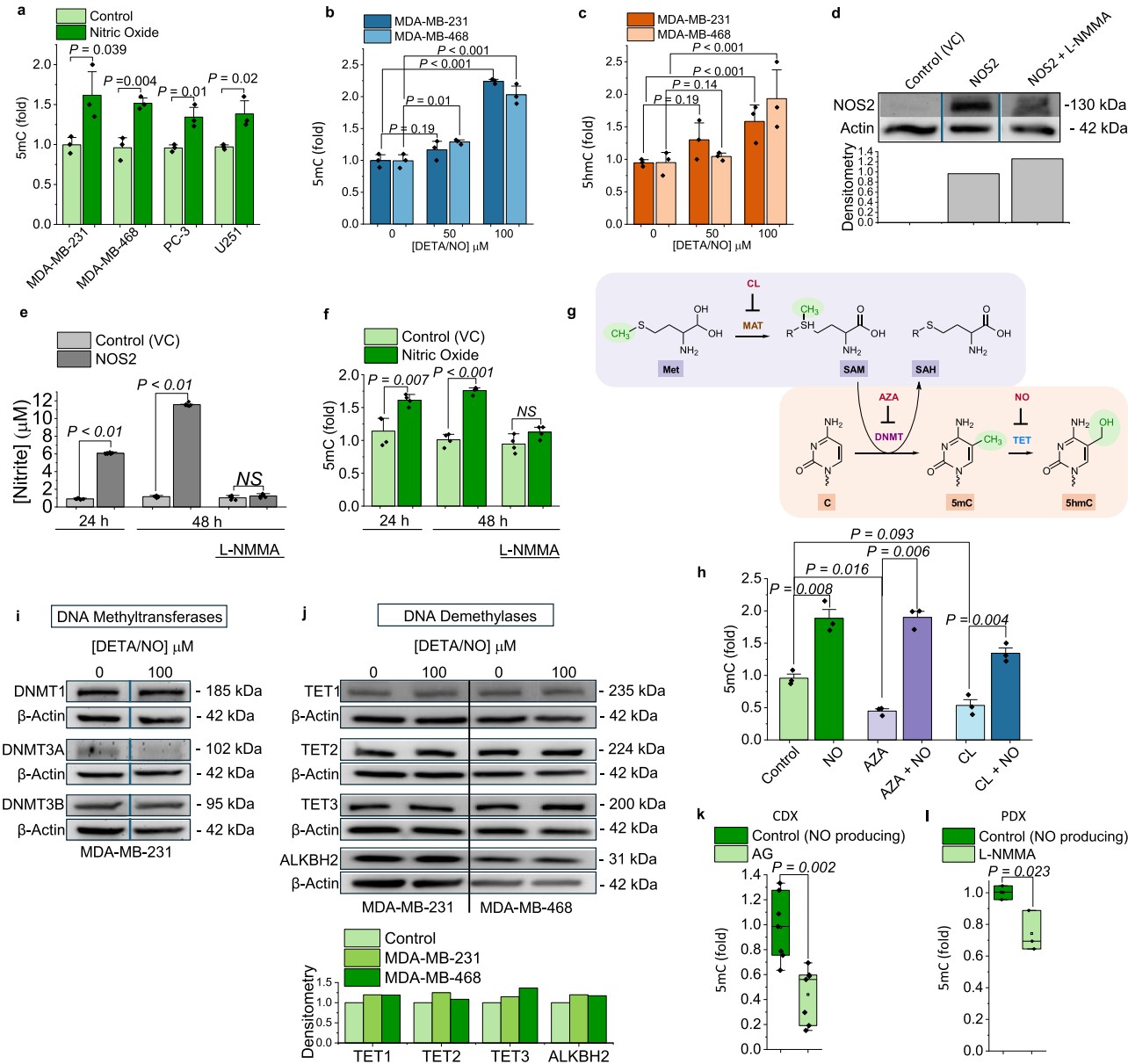

**Fig. 3 | NO increases 5mC on DNA from cancer cells and from tumors. a** Relative abundance of 5mC-DNA in cancer cells treated without (light bars) or with NO (dark bars, DETA/NO 100 µM, 24 h). **b** Relative abundance of 5mC-DNA and (**c**) 5hmC-DNA from TNBC cells that were chronically treated with NO for 10 days (DETA/NO 50 and 100 µM added every 48 h), (**a**–**c**, ELISA, mean ± SEM, n = 3). **d** Immunoblot for NOS2 protein in MDA-MB-231 empty vector control (VC), MDA-MB-231 NOS2-expressing, and MDA-MB-231 NOS2 + L-NMMA (1 mM), at 24 h. **e** Total NO synthesis as measured by nitrite, n = 6, mean ± SD), and (**f**) 5mC-DNA (ELISA) from MDA-MB-231 (VC) and MDA-MB-231 NOS2 +/- L-NMMA cells cultured for 24 and 48 hours, (n = 4, mean ± SD). **g** How inhibitors cycloleucine (CL), 5-Azacytidine (AZA), and NO interact with DNMT, MAT (methionine adenosyltransferase), and TET to affect 5mC-, 5hmC-DNA levels. **h** 5mC-DNA from MDA-MB-231 cells treated for 24 h +/- NO (100 µM DETA/NO, green bar), or +/- 1 mM AZA (purple bars), or +/- 1 mM CL (blue bars), ELISA, (n = 3, mean ± SD). Representative immunoblot and densitometry for

(**i**) DNMT1, 3A, and 3B and (**j**) TET1, 2, 3, and ALKBH2 from MDA-MB-231 and MDA-MB-468 cells cultured for 10 days with NO (DETA/NO, 100 µM), n = 3 separate experiments. **k** 5mC-DNA levels from mice with NO-producing MDA-MB-231 xenograft tumors (dark green bars), and from tumors that did not synthesize NO (treated aminoguanidine (AG), light bars), ELISA, n = 7/group. **l** 5mC-DNA from mice with TNBC PDX tumors that did synthesize NO (dark green bars) and from tumors that did not synthesize NO (L-NMMA), ELISA, n = 3/group. Data are presented as box plots (center line at the median, upper and lower bounds are 75th and 25th percentile with whisker at 1.5 IQR. Each dot represents one mouse. For bar graphs data are mean ± S.E.M or ± SD, *P*-values were determined by unpaired two-tailed Student's t-tests, *NS* = not significant, blue vertical line on immunoblots indicates splicing of lanes that were run on the same gel but were noncontiguous. Source data are provided as a Source Data file.

DNMT3a, DNMT3b) and DNA demethylases (TET1,2,3, and ALKBH2) (Fig. 3i, j). For all enzymes, there was almost no change in protein expression in response to NO after 10 days. Together these data further support the hypothesis that NO-mediated increases in 5mC result from inhibition of TET demethylases and are not a result of changes in the expression levels of DNA methyl-modifying enzymes or increased DNA methyltransferase activity.

## Nitric oxide increases 5mC in DNA from tumors in vivo
To investigate whether NO could increase 5mC in vivo we used a mouse xenograft model of *NOS2*-expressing cell-line derived tumors. Mice bearing *NOS2*-expressing MDA-MB-231 xenograft tumors were divided into two groups; half were treated with aminoguanidine (AG), a selective inhibitor of NOS2, and the other half were treated with saline (control). After 37 days of treatment, the

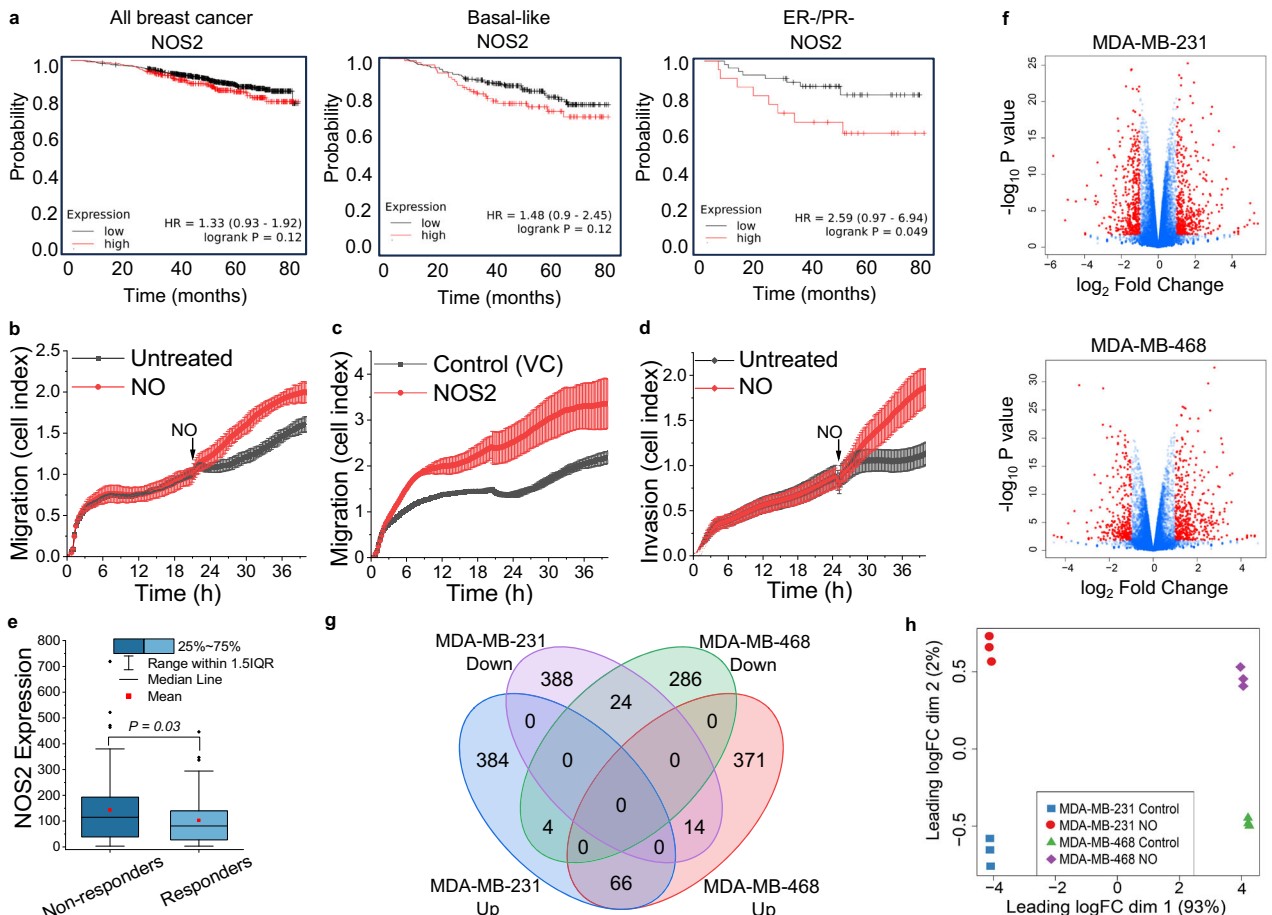

**Fig. 4 | NO is associated with aggressive cancer phenotypes and causes transcriptional changes in breast cancer. a** Patient survival as a function of NOS2 gene expression and hormone receptor status using Kaplan-Meier Plotter (public database includes 7830 unique breast cancer samples). (**b**–**d**) Cell migration and invasion measured in real time by the xCELLigence® DP system. **b** MDA-MB-231 cells were plated and allowed to adhere for 20 h before the addition of NO (100 μM DETA/NO). **c** MDA-MB-231 empty vector control (VC) and MDA-MB-231 NOS2-expressing cells were plated, and migration was monitored for 40 h. **d** Invasion: MDA-MB-231 cells were plated on a Matrigel® matrix and incubated for 20 h before the addition of NO (100 μM DETA/NO), (**b**–**d**, data lines represent the mean +/- SD, n = 2). **e** NOS2 gene expression in tumors from TNBC patients who responded to chemotherapy and patients who did not respond to chemotherapy (n = 164). The two cohorts were compared using ROC Plotter platform, Mann–Whitney test. (**f**–**h**) mRNA-Seq was conducted on samples from MDA-MB-231 and MDA-MB-468 cells treated with or without NO (DETA/NO 100 μM; 10 days, n = 3 biological replicates for each cell type). **f** Volcano plots of NO-mediated mRNA changes. Significantly up- and down-regulated genes are indicated in red, (significance indicated as log$_2$ fold > 1 or <−1 and -log$_{10}$ *P*-values > 0.05) Differential expression analysis was completed using quasi-likelihood F-tests (glmQLFTest function) in edgeR v3.40.2. P-values from all tests were adjusted to generate FDR (False Discovery Rate) with built-in Benjamini-Hochberg method. **g** Venn diagram demonstrates the number of significant NO-regulated genes that overlap between the two cell types. **h** Multidimensional scaling (MDS) analysis of gene expression from mRNA-Seq data sets. Source data are provided as a Source Data file.

tumors were removed, the DNA extracted, and 5mC-DNA was quantified (Fig. 3k). 5mC in DNA from the NO-producing tumors was significantly greater than in the tumors where NO synthesis was inhibited. As further confirmation that NO regulates 5mC in vivo, we measured 5mC-DNA in NOS2-positive patient derived xenograft (PDX) tumors (Fig. 3l). In this experiment the control group received a vehicle saline injection, and the treatment group was administered the pan-NOS inhibitor NG-monomethyl-L-arginine (L-NMMA) daily. After 40 days the tumors were excised and 5mC was measured. Again, in the NO-producing PDX tumors, 5mC was significantly higher than in the tumors where NO synthesis was inhibited (Fig. 3l).

### NOS2 expression and NO production drive aggressive cancer phenotypes

Clinically, *NOS2* expression in tumors is associated with worse patient outcomes and poor responses to therapy. We used Kaplan-Meier Plotter[60] to analyze transcriptomic datasets of metastatic breast cancer patients found in GEO, EGA and TCGA. When we examined NOS2 expression in 3 breast cancer patient groups (all subtypes, basal-like, and ER⁻/PR⁻) we found that high *NOS2* expression was associated with decreased overall survival (confirming previous reports[27]) (Fig. 4a). To experimentally determine whether NO would result in a more aggressive cell phenotype in vitro, we measured cell migration and invasion (two hallmarks of cancer) in real time of TNBC cells exposed to NO. In both cells exposed to exogenous NO and in cells endogenously synthesizing NO, the rates of cell migration and invasion were increased compared to untreated control cells (Fig. 4b–d), consistent with what we and others have shown previously[22,61]. Another phenotype associated with tumor aggressiveness is resistance to chemotherapy. Using the ROC plotter platform[62], we analyzed the expression levels of *NOS2* as a predictive biomarker of efficacy of any chemotherapy in TNBC patients (n = 164; response based on relapse-free survival at 5 years). Patient non-responders to therapy had significantly higher *NOS2* expression than patient responders to therapy (Fig. 4e).

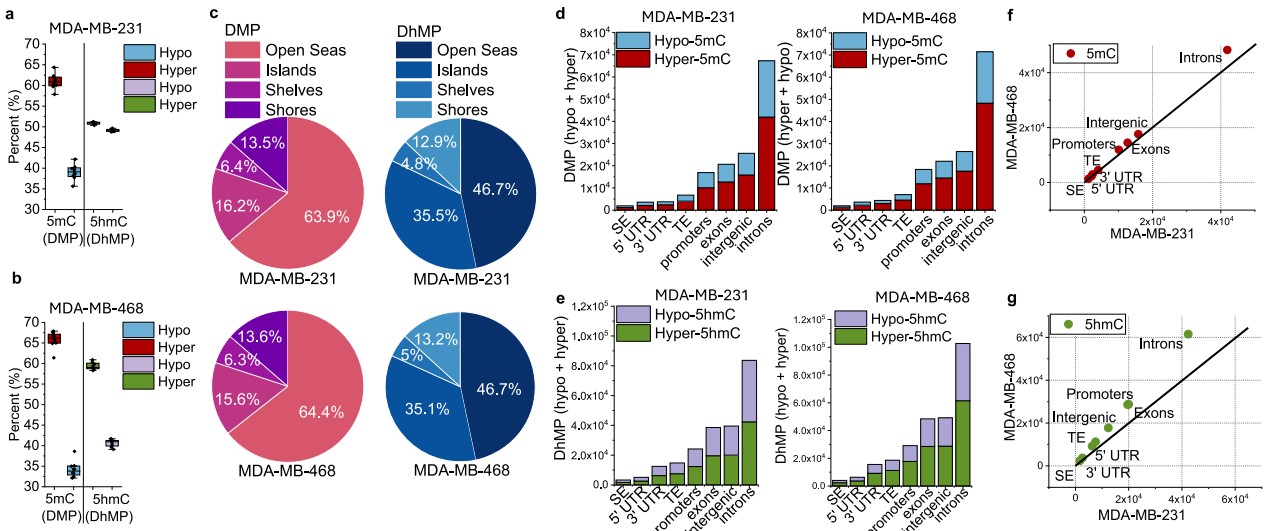

**Fig. 5 | NO increases 5mC and 5hmC in DNA at specific genomic locations.** Oxidative reduced representation bisulfite sequencing (oxRRBS) was conducted on samples from MDA-MB-231 and MDA-MB-468 cells treated with or without NO (DETA/NO 100 μM; 10 days, n = 2 biological replicates). **a, b** percent change of hyper- and hypo- differentially methylated positions (DMP) and hyper- and hypo- differentially hydroxymethylated positions (DhMP) across all 8 annotated sites in both cell types, each dot represents one annotated site, center line at the median, upper bound at 75th percentile, lower bound at 25th percentile. **c** Annotated CpG sites of NO-mediated hyper-DMP and hyper-DhMP (DMP and DhMP = *P-value* < 0.05, mean difference in abs (beta value) of ≥ 0.1 according to RnBeads). **d** NO-mediated hyper- and hypo-differentially methylated CpG positions and **e** hyper- and hypo-differentially hydroxymethylated CpG positions at functional elements (5'UTR = 5'untranslated region (5'UTR), 3'UTR = 3'untranslated region (5'UTR), SE = Super enhancers, and TE = typical enhancers. Comparing differences in the magnitude of (**f**) DMP and (**g**) DhMP at functional elements between both cell types. Source data are provided as a Source Data file.

## Nitric oxide induces extensive transcriptional changes in TNBC cells

With the observation that NO directly inhibits TET demethylase activity leading to global increases in 5mC, we sought to decipher whether this was functionally associated with transcriptional changes in NO-regulated genes that may drive aggressive phenotypes. First, we quantified changes in transcription by performing RNA-sequencing on samples from two TNBC cell lines treated chronically (10 days) with NO. In both cell lines, NO significantly up- and down-regulated several hundred genes compared to untreated control cells (Fig. 4f, g). There were 880 significantly differentially expressed genes in the MDA-MB-231 cells with FDR (False Discovery Rate) < 0.05 (454 upregulated log₂FC > 1, 426 downregulated log₂FC < −1) and 765 significantly differentially expressed genes in the MDA-MB-468 cells (451 upregulated log₂FC > 1314 downregulated log₂FC < −1). There was only a 12–14% overlap in common genes transcriptionally regulated by NO between the two cell types (90 in total), which is likely due to significant differences in their basal transcriptional profiles (control cells not treated with NO). As illustrated by multidimensional scaling (MDS) analysis the expression profiles differed far more between the two different untreated cell types than between the NO treated and control sample of the same cell type (Fig. 4h). Despite only modest overlap in specific genes transcriptionally regulated by NO in both cell types, Gene Set Enrichment Analysis (GSEA) of the 90 genes differentially expressed in the same direction identified several KEGG pathways relevant to cancer progression (Supplementary Table 1).

## Nitric oxide increases 5mC and 5hmC differentially at specific genomic features

To determine if NO-mediated changes in 5mC/5hmC were functionally associated with changes in gene expression, we identified the locations of 5mC/5hmC on a genome-wide scale at single-nucleotide resolution by performing oxidative reduced representation bisulfite sequencing (oxRRBS) on samples from two TNBC cell lines chronically treated with NO for 10 days. Between the NO-treated cells and the untreated cells we identified differentially methylated positions (DMPs, 5mC) and differentially hydroxymethylated positions (DhMPs, 5hmC), defined as: p < 0.05 and difference in β-value > 0.1. Although we measured both increases and decreases in 5mC and 5hmC, there was a net overall increase in both 5mC and 5hmC in the NO-treated cells compared to the untreated control cells (Fig. 5a, b). This was consistent with the data in Fig. 3 where global increases in both 5mC and 5hmC were measured by ELISA assay. When we examined annotated CpG sites (islands, shores, shelves, and open seas)[63] we found that the majority (>60%) of DMPs and (>45%) DhMPs occurred at open sea positions (Fig. 5c). We then focused on specific functional elements (super enhancers (SE), 5' UTR, 3' UTR, typical enhancers (TE), promoters, exons, intergenic regions, and introns) to determine if they also exhibited specific patterns of 5mC/5hmC enrichment. We identified hyper- and hypo-DMPs and DhMPs at all genic annotations (Fig. 5d, e). The distribution and magnitude of DMPs and DhMPs was similar across both cell lines with the majority located at introns, followed by intergenic regions and exons (Fig. 5f, g).

## Determination of 5mC- and 5hmC-associated transcriptional changes

Having demonstrated that NO increased 5mC/5hmC in DNA at gene-regulatory genomic locations, and that NO produced significant transcriptional changes, we attempted to link changes in DNA methylation to the changes in gene expression. We identified overlaps between significantly expressed genes (RNA-seq) and their β-values (the degree of CpG methylation, oxRRBS) at that gene or at gene regulatory loci associated with that gene (i.e., promoters, gene bodies, enhancers, super enhancers). For both control and NO treatment groups, there was a clear negative correlation between 5mC β-values at promoters and gene expression in both cell types (Supplementary Fig. 3). We next identified specific genes that had significant transcriptional changes in response to NO (upregulated or downregulated) and also had significant changes in methylation (increase or decrease in β-value for 5mC or 5hmC) at specific gene-regulatory genomic loci (Fig. 6a, b,

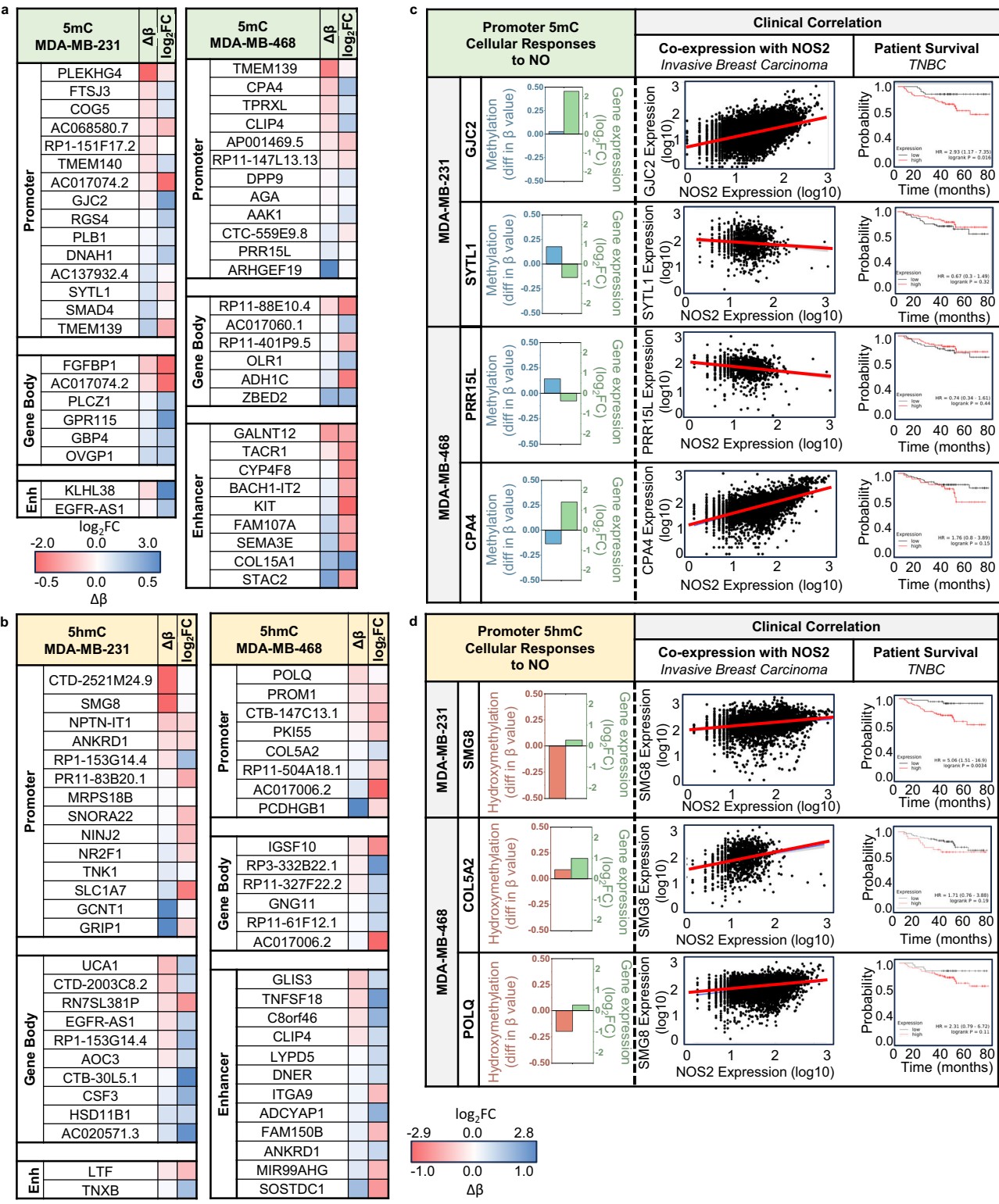

**Fig. 6 | NO-dependent changes in 5mC and 5hmC are associated with transcriptional changes in NO-responsive genes. a, b** oxRRBS data from MDA-MB-231 and MDA-MB-468 cells treated with or without NO for 10 days. β-values (5mC or 5hmC) at Enhancers, Gene Bodies, and Promoters (as determined by RnBeads) are ranked from lowest to highest and are paired with the expression changes (RNA-seq) in their associated genes. **a** Changes in gene expression and in 5mC in DNA from NO-treated MDA-MB-231 and MDA-MB-468 cells. **b** Changes in gene expression and in 5hmC in DNA from NO-treated MDA-MB-231 and MDA-MB-468 cells. Specific tumor-permissive genes that are associated with 5mC (**c**) and with 5hmC

(**d**) at their promoter regions. Left panels Cellular Responses to NO, are representative genes shown with their difference in β-value at their promoter region (oxRRBS data) and their expression level (mRNA-seq) in response to NO. Right side, Clinical Correlations, are scatter plots demonstrating correlations between NOS2 expression and the indicated gene in tumors from patients with aggressive breast cancer. Far right column, Patient Survival TNBC, are Kaplan-Meier plots of TNBC patient survival and the expression levels high (red line) or low (black line) of the indicated gene.

Supplementary Tables 2, 3). Although the correlations between methylation status (hyper or hypo, 5mC or 5hmC) of specific gene-regulatory regions and the direction of transcriptional changes were not mutually exclusive, certain trends did emerge. For example, increases of 5mC/5hmC at promoters were more associated with downregulated genes whereas gene body enrichment of 5mC/5hmC was more associated with upregulated genes (Supplementary Fig. 4). Looking at typical enhancers, increased 5mC/5hmC correlated with downregulation of associated genes. While the connections between many of the differentially expressed genes (Fig. 6) and cancer are either unknown or have not been fully established, some of these genes have been experimentally or clinically linked to breast cancer progression (Fig. 6c, d). On the left sides of Fig. 6 c, d (Cellular Responses to NO) are select genes that are transcriptionally regulated by NO and show significant changes in their β-values at the promoter regions for these genes (Fig. 6c for 5mC, 6d for 5hmC). On the right side of these figures (Clinical Correlation) are results from analysis of publicly available data sets of gene expression from tumors of patients with aggressive breast cancers[64]. Genes that were transcriptionally regulated by NO in our cell culture models also had directionally similar gene expression changes in *NOS2*-expressing patient tumors. Kaplan-Meier plots demonstrate the correlation between the NO-regulated gene and patient survival[60]. Although a direct relationship between these NO-regulated genes and cancer progression have yet to be documented in the scientific literature, some reports demonstrate that the genes upregulated by NO (*GJC2, CPA4, SMG8, COL5A2, POLQ*)[65–69] and the genes downregulated by NO (*SYTL1, PRR15L*)[70,71] are associated with deleterious outcomes when up- or down-regulated in cancer patients. These data demonstrate that genes that exhibit changes in their promoter 5mC/5hmC status and are transcriptionally regulated by NO in breast cancer cells in vitro show similar trends in vivo in patient tumors, suggesting a link between NO-regulated genes and poor patient outcomes in breast cancer.

## Discussion

Here we demonstrate that physiologic NO concentrations directly inhibit TET and ALKBH2 via a unique mechanism involving DNIC assembly at the catalytic non-heme iron atom. Moreover, we find that in cancer cells exposed to exogenous NO, or in cells endogenously synthesizing NO, 5mC/5hmC in DNA is increased, and that 5mC and 5hmC are enriched on gene-regulatory loci (i.e., promoter, gene body, enhancer regions) of specific NO-regulated genes.

There have been several excellent studies that provide insight into mechanisms through which NO influences DNA methylation patterns[72], predominantly by modulating the expression levels of DNA methyl-modifying enzymes (i.e., methyltransferases or demethylases). For example, Switzer et al. demonstrated that NO caused a decrease in DNA methylation via *S*-nitrosation of DNA methyltransferase 1 (DNMT1) which induced DNMT1 degradation[73]. Similarly, another study demonstrated that, in the vascular wall, NO led to an overall decrease in DNA methylation by increasing TET activity and decreasing DNMT activity[74]. Conversely, others have shown that NO enhances the enzymatic activity of DNA methyltransferases (DNMTs) and down-regulates TET enzymes to cause an increase in DNA methylation[75]. While our study does not refute these previous findings, we provide multiple lines of biochemical evidence to demonstrate that NO can directly inhibit the catalytic activities of TET enzymes leading to increased 5mC/5hmC on DNA.

There are no known structures of NO bound to TET/ALKBH2 and very little data on NO bound to any member of this family of proteins with the exception of two examples of the crystalline form demonstrating formation of the mono-NO at the iron site[76,77]. The non-heme Fe(II) in TET enzymes binds 2-OG in a bidentate manner during the catalytic cycle which leaves one coordination site available for $O_2$ binding, or as we initially hypothesized, for NO binding. Intriguingly,

EPR and modeling studies demonstrated that NO preferentially formed a DNIC and not a mononitrosyl, suggesting NO binds to the enzyme in its resting state prior to 2-OG coordination. This means that NO may compete with 2-OG and not $O_2$ to prevent the interaction of TET with all its DNA substrates allowing their differential accumulation. Kinetic experiments, conducted at room air oxygen concentrations (-220 μM), revealed that low physiologic concentrations of NO (nM in the case of ALKBH2) were sufficient to inhibit demethylase activity in vitro and in vivo. This indicates that TET enzymes have a much higher affinity for NO than $O_2$. We also found that NO significantly inhibited TET's initial oxidation step (5mC ⇒ 5hmC) more potently than its subsequent oxidation steps (5hmC ⇒ 5fC ⇒ 5caC). Although further mechanistic studies are needed to delineate factors contributing to the differential buildup of TET substrates, these differences may partially be a function of differences in the catalytic rates for each of its oxidation steps, with the first step, conversion of 5mC ⇒ 5hmC, being the fastest[78].

In cells, most studies by other groups have shown that small molecule chemical inhibitors of TET enzymes cause global increases in 5mC and decreases in 5hmC[79,80]. However, other labs have reported that TET inhibitors[81], *TET* knock down[82], or decreased substrate availability (hypoxia, $O_2$)[83,84] can lead to an increase in cellular 5hmC. Our cellular studies demonstrated that NO caused global, and loci-specific, increases in both 5mC and 5hmC in DNA. It is intuitive that 5mC would increase consequent to TET inhibition, yet mechanisms to explain why 5hmC also increases are less obvious. It is possible that of the 3 TET paralogs that are expressed in the cells we tested, one of the paralogs may be more sensitive to inhibition by NO, while one of the other ones may have a greater affinity for 5mC than 5hmC. Insight into these questions comes from a recent paper by Belle et. al.[81] where they examined the sensitivities of human TET enzymes to various inhibitors. They found that most inhibitors manifested similar potencies for all TET paralogs and caused an increase in 5hmC in isolated enzymes and in cells. Another group found similar results using an embryonic carcinoma cell model where TET2,3 depletion caused increased 5hmC[82]. An alternate explanation for NO-mediated increases in 5hmC can be learned from studies on hypoxia. Oxygen ($O_2$) is a substrate for TET enzymes, and therefore TET enzymes are known to be catalytically inactive in physiologically hypoxic environments (below the $K_M$ for $O_2$). Despite this fact, numerous reports have demonstrated that hypoxia increases 5hmC levels in cells, which, similar to our data, seems counterintuitive. One of the initial reports of this phenomenon in 2014 demonstrated that hypoxia increased global 5hmC in DNA that accumulated at canonical hypoxia response genes[84]. Since then, numerous studies have replicated these findings in various model systems and demonstrated that both short (24 h) and long-term (10 day) hypoxia increases 5hmC in DNA[83,85–88]. One explanation cited for increased 5hmC under hypoxia was that hypoxia increased the expression levels of some *TET* paralogs. We explored this possibility and determined that concentrations of NO that increased 5hmC did not change the expression levels of *TET1, 2*, or *3*. This is consistent with other studies demonstrating that expression profiles of the TETs do not mirror dynamic changes in global 5hmC levels[89,90]. In another study, researchers treated cells with cobalt chloride ($CoCl_2$), a chemical hypoxia mimetic that targets many of the same metalloproteins as NO. Consistent with our results, they measured a significant increase in 5hmC under normoxia (21% $O_2$)[91].

Considering other explanations for increases in 5hmC, it has been shown that the HDAC inhibitor panobinostat increased 5hmC levels in cells in a concentration- and TET1-activity-dependent manner[81]. The group theorized that HDAC inhibition, which results in elevated levels of histone lysine acetylation, may improve the accessibility of DNA to TET enzymes as well as increase *TET* expression. This is potentially relevant to our data as we have demonstrated that physiologic NO concentrations can regulate histone post-translational modifications

in TNBC cells, including increasing acetylation at specific lysine residues (i.e., H3K27)[42,43,61]. Although we favor a direct mechanism of TET inhibition by NO to explain the majority of increases in 5hmC, we cannot fully rule out other concurrent mechanisms.

Regardless of the mechanism of TET inhibition, in vitro (cell culture) and in vivo (cell-derived and patient derived xenograft) models demonstrated that NO was associated with global increases in 5mC and 5hmC, which is consistent with other reports demonstrating that *TET2,3* knockdown in cancer cells resulted in an increase in 5mC and 5hmC[82]. Moreover, since TET inhibition by NO is reversible this suggests that NO could act as a molecular switch to turn on and off demethylase activity as a mechanism of dynamic regulation of DNA methylation. Cellular expression of *NOS* paralogs, which synthesize NO at different rates, could be the key determinant of whether NO signals epigenetically, through changes in DNA methylation, or predominantly through canonical mechanisms.

These studies investigated transcriptional changes in cells exposed to chronic (10-day) physiological, low steady-state NO concentrations, an approach that more closely reflects the cellular microenvironment of *NOS2*-expressing tumors where NO synthesis is constitutively active. Although we measured thousands of genes that were transcriptionally changing in response to NO, the overlap between specific genes in two transcriptionally diverse cell types (MDA-MB-231, -468) was relatively small. A major question is whether TET inhibition by NO to increase 5mC/5hmC is a specific mechanism to mediate these transcriptional changes or whether this represents a more generalized phenomenon associated with dysregulated gene expression. The fact that not all CpG sites are increasing 5mC/5hmC equally in response to NO suggests that there are hot-spots of TET activity that may reflect TET's genomic location. We suspect that the sites of NO-dependent CpG methylation correspond to TET target regions of DNA in open chromatin where transcriptional activity is occurring or at regulatory regions that control transcriptional activity. In this regard, our ability to link promoter or gene body methylation (5mC or 5hmC) to the regulation of specific genes did not prove to be 100% consistent across genomic regions or cell lines. For example, in both cell types NO-regulated genes were associated with increases or decreases in promoter 5mC and 5hmC. 5mC methylation was increased at the promoter of *GJC2* which was transcriptionally upregulated by NO. Yet, 5mC methylation was also increased at the promoter regions of *SYTL1* and *PRR15L*, which were transcriptionally downregulated by NO. At gene bodies, 5mC was largely associated with increased gene expression. At transcription start sites (TSS) 5hmC is usually present at genes with high CpG promoters, marked by bivalent histone modifications, suggesting a role in regulating gene expression by modulating chromatin accessibility or inhibiting repressor binding[92]. We found that loss of 5hmC correlated to increased gene expression for *POLQ* and *SMG8*, whereas gain of 5hmC correlated to increased expression of *COL5A2*.

Although linking 5mC/5hmC to changes in gene expression in our models proved to be complicated, we did find that many of the genes transcriptionally regulated by NO in cell culture showed similar directional expression changes in hundreds of clinical samples of *NOS2*-expressing tumors. Moreover, the directional changes in these NO-regulated methylated genes correlated to decreased patient survival for patients harboring *NOS2*-expressing tumors. For decades it has been widely accepted that DNA methylation at CpG promoter regions is associated with transcriptional repression. Yet, numerous recent studies challenge this by demonstrating the persistent inability to attribute gene expression causality to methylation at specific DNA loci[93]. Several studies have shown an inverse correlation between DNA methylation of the first intron and gene expression across multiple tissues and species[94]. Numerous other studies have demonstrated that gain of promoter methylation is associated with increased gene expression[95–97]. For example, a large study of prostate cancers (1117 samples) found that hypermethylated genes were strongly associated with increased gene expression[98]. Further complicating the issue are the multiple studies demonstrating that both transcription factor binding and changes in gene expression can in some cases precede DNA demethylation[99,100]. DNA methylation appears to be highly context-specific, even in a single cancer type like TNBC. The correlations between *TET* expression, DNA methylation, and tumor progression are highly dependent on both the TET paralogs (1, 2, or 3) and the type of breast cancer. For example, *TET1* expression and 5hmC levels are decreased in tumor samples from luminal A, luminal B, and HER2-positive subtypes compared to normal breast tissue samples. Low *TET1* expression and 5hmC levels also correlate with larger tumors, advanced stage, lymph node involvement, and poor patient survival[101–104]. Conversely, in TNBC, *TET1* expression is increased in tumor tissue samples compared to normal breast tissue samples, and its high expression correlates with poor patient outcomes[105,106]. Taken together, we favor a more stochastic view of NO-mediated gene expression changes where NO inhibition of TET may predominantly lead to indiscriminate disruption of DNA methylation that cancers exploit to regulate tumor-permissive gene expression programs. This may ultimately benefit cancer cell survival by creating transcriptional diversity among populations of cells independent of their genetic and transcriptomic background. This theory is supported by our data demonstrating that although NO-dependent transcriptomic changes were vastly different between two TNBC cell lines, similarities were noted at the level of tumor pathway analysis.

Lastly, in these studies, we mainly focused on the functional consequences of TET inhibition by NO. However, our data also demonstrated that the DNA repair enzyme ALKBH2 was inhibited by NO, which may have implications for cancer etiology and treatment. By inhibiting ALKBH2, NO could result in the accumulation of 1-methyladenine and 3-methylcytosine which are major toxic and mutagenic DNA lesions, thereby leading to loss of genome integrity and carcinogenesis. Conversely, our findings may provide a rationale for the use of therapeutic alkylating agents in *NOS2*-expressing tumors. In this case, endogenous NO synthesis could sensitize the tumors to alkylating agents by increasing alkylation-mediated DNA damage and cell death.

In conclusion, this study demonstrates that NO is an endogenous regulator of TET activity and DNA methylation. This represents an unprecedented functional role for NO in regulating steady-state DNA methylation (and hydroxymethylation) levels. How changes in DNA 5mC/5hmC at specific loci regulate the expression of NO-responsive genes and how this mechanism synergizes with or antagonizes other canonical NO signaling mechanisms is still an open question. In cancer, further mechanistic studies are needed to fully understand the functional consequences of NO-mediated TET inhibition in relation to transcriptional malleability, transcriptional heterogeneity, and phenotypic plasticity—all associated with more aggressive tumors, worse patient prognosis, and resistance to chemotherapies. Although the findings presented herein have been in the context of cancers (breast), we suspect that the fundamental discovery that NO inhibits TET enzymes to change DNA methylation patterns is a contributing factor to numerous diseases where there is dysregulated NO synthesis and aberrant DNA methylation patterns[107]. Moreover, this discovery raises the possibility that NO could regulate DNA methylation to control gene expression under physiological settings which should be explored further. Our previous work demonstrated that NO is an endogenous regulator of histone post-translational modifications[42,43] and mRNA methylation[44], and here we show how NO regulates DNA methylation. Therefore, in addition to its canonical roles in cell signaling and gene expression, NO should be recognized as a dominant regulator of the epigenetic landscape[108].

## Methods

### Ethical approval

This study complies with all relevant ethical regulations and ethics approval for all animal studies was approved. Those conducted by the National Cancer Institute were approved by The Institutional Animal Care and Use Committee (IACUC) under the Animal Study Protocol (#21-109), and those conducted at the Houston Methodist Hospital Research Institute were approved by the Animal Care and Review Office (IACUC number IS00007220).

### Statistics and reproducibility

No statistical method was used to predetermine sample size. No data were excluded from the analyses. The experiments were not randomized. The Investigators were not blinded to allocation during experiments and outcome assessment. For cell culture experiments, at a minimum, all data were collected in technical and biological triplicates to facilitate statistical comparisons between groups. The statistical test used, error bars, and $n$ values are defined in each figure legend. P-values were calculated in OriginPro. For the RNA-sequencing experiments, $n = 3$ was used and for oxRRBS DNA sequencing $n = 2$ was used. The results were similar between biological replicates.

### Chemicals

(Z)-1-[N-(2-aminoethyl)-N-(2-ammonioethyl)amino]diazen-1-ium-1,2-diolate (DETA/NO), (Z)-1-[N-[3-aminopropyl]-N-[4-(3-aminopropy-lammonio)butyl]-amino]diazen-1-ium-1,2-diolate (SPER/NO), 2-(N,N-Diethylamino)-diazenolate-2-oxide (Diethylamine nonoate) (DEA/NO) were each a gift from Dr. Joseph E. Saavedra (NCI). Sulfanilamide (SULF; prepared to 2% (w/v) in 5% HCl and filtered to remove trace particles), N-(1-naphthyl)ethylenediamine dihydrochloride (NEDD) (Riedel-de Haen, Germany) and prepared to 0.1% (w/v) in $H_2O$ and filtered to remove trace particles. Ammonium iron(II) sulfate hexahydrate (($NH_4$)$_2$Fe(SO$_4$)$_2$·6$H_2O$; purity 99.997%), α-ketoglutaric acid sodium salt (αKG (2-OG); purity >98%), (+)-sodium L-ascorbate (ascorbic acid; purity >98%), NaCl (purity ≥99%), aminoguanidine hydrochloride (purity ≥98%), 5-Azacytidine (AZA; purity ≥98%), 3-Hydroxypicolinic acid (3-HPA; purity ≥99%),ammonium citrate dibasic (purity ≥99%), bovine serum albumin (BSA), catalase (bovine liver) all obtained from Sigma-Aldrich. Dithiothreitol (DTT) (Bio-Rad). HEPES buffer (pH 7.2−7.5) (Gibco). Adenosine 5′-triphosphate disodium salt hydrate (Thermo Scientific). Cycloleucine (purity >98%) (Chem Impex). N(G)-nitro-L-arginine methyl ester hydrochloride (L-NAME; purity >99%) (Cayman Chemical). Amlodipine besylate (Major Pharmaceuticals). L-NG-Monomethylarginine, Acetate Salt (L-NMMA; purity ≥99%) (Santa Cruz Biotechnology).

### Cell culture and DETA/NO treatment

Both MDA-MB-231 and MDA-MB-468 cells were obtained from American Type Culture Collection (ATCC) and cultured in high glucose DMEM (Gibco #11995-065) supplemented with 10% fetal bovine serum (Gibco #26140-079) and 1% penicillin/streptomycin (Gibco #15140122) at 37 °C in a humified atmosphere of 5% $CO_2$. Cell line authentication was performed using short tandem repeat analysis. MDA-MB-231 cells expressing NOS2 were graciously provided by Dr. Sharon Glynn of the University of Galway and were stably transduced with either the NOS2 lentiviral vector obtained from Origene or an empty vector. The cells were cultured in RPMI 1640 with 5% L-glutamine (Millipore Sigma #R8758), 10% fetal bovine serum (FBS; Gibco), 1% penicillin/streptomycin (P/S; Gibco), selected with 2 μg/mL puromycin at 37 °C with 5% $CO_2$. To treat cells with the NO donor DETA/NO, DETA/NO was first thawed on ice and diluted 1:1000 in 10 mM NaOH before measuring absorbance at λ = 250 nm to determine the concentration of the stock solution. The same procedure was used for NO donors SPER/NO and DEA/NO. For the low dose, chronic treatment, cells were plated in a

10 cm dish and treated with 50 μM or 100 μM DETA/NO for a total of 10 days, while passaging and re-treating every 2 days.

### Griess assay for nitrite quantification

The Griess assay for nitrite detection was conducted on media samples from cultured cells. In each well of a 96-well plate, 100 μL of media was introduced, after which 50 μL of 2% (w/v) sulfanilamide in 5% HCl and 50 μL of 0.1% N-(1-Naphthyl)ethylenediamine dihydrochloride (NEDD) in $H_2O$ were added to the media. Plates were then incubated at 37 °C for 40 min. The absorbance of the samples was measured at 540 nm using a BioTek Synergy HTX Multimode Reader and nitrite concentration was calculated using linear regression of sodium nitrite standards.

### Cell proliferation and invasion xCELLigence assays

Cell proliferation and invasion was measured using the Agilent xCEL-Ligence Real-Time Cell Analysis system. For proliferation studies, 7500 cells/well were plated onto e-plates and placed in the 37 °C, 5% $CO_2$ system to allow cell adherence. After 21 h, if cells were treated, they received 100 μM DETA/NO and were placed back into the system for continued observation for 24 h. For invasion studies, 50 μL Matrigel was first poured onto the e-plates. After allowing the Matrigel to harden, 20,000 cells/well were plated and allowed to adhere. After 25 h, if cells were treated, they received 100 μM DETA/NO and were placed back into the system for continued observation for 24 h.

### Mouse xenograft studies

Animals were housed at constant room temperature (22 ± 1 °C) and relative humidity (55 ± 5%) under a 12 h light/dark cycle (lights on from 7:00 AM to 7:00 PM). Food and water were available ad libitum. Mice were always tested during the light phase. According to the institutional review boards the maximal size permitted for any single tumor is 2000 mm3, this was not exceeded for any mouse in either xenograft study.

### Cell-derived xenograft mouse studies

Mouse studies were performed at the National Cancer Institute. Eight-week-old female Balb/c athymic nude mice were provided from the Frederick Cancer Research and Development Center Animal Production Area (Frederick, MD). Only female mice were used as TNBC almost exclusively occurs in women. The Institutional Animal Care and Use Committee (IACUC) approved the Animal Study Protocol (#21-109). Each mouse received a mammary fat pad injection of 750,000 MDA-MB-231 human triple-negative breast cancer (TNBC) cells overexpressing GFP. Following one week of tumor growth, mice were randomly assigned to control and treatment groups. The treatment group received the NOS inhibitor aminoguanidine at a concentration of 0.5 g/L in filter-sterilized drinking water. After 6 weeks, the mice were sacrificed and subjected to imaging. All animal protocols were approved and followed the principles outlined in the Guide for the Care and Use of Laboratory Animals by the Institute of Laboratory Animal Resources, National Research Council.

### Patient-derived xenograft mouse studies

All animal procedures were approved by the Houston Methodist Hospital Research Institute Animal Care and Review Office (IACUC number IS00007220). We did not perform a clinical trial in this present study. PDX in vivo studies were conducted by the Chang Lab at Houston Methodist Hospital on three TNBC PDXs: BCM-5998, BCM-3107, and BCM-3807. These samples were part of a previous study[109]. Written informed consent to donate samples for PDX research was obtained from all patients before sample and data collection under protocol IRB # Pro00012329 and approved by the Institutional Review Board of Houston Methodist Hospital. PDXs were transplanted into the

**Table 1 | List of antibodies used in the study**

| Target protein | Concentration | Source/Isotype | Catalog Number |
|---|---|---|---|
| TET1 | 1:1000 | Rabbit | GeneTex GTX124207 |
| TET2 | 1.5 μg/mL | Mouse | Active Motif 61390 |
| TET3 | 1:1000 | Mouse | Abiocode M1092-3 |
| ALKBH2 | 1:4000 | Rabbit | Abcam ab154859 |
| DNMT1 | 1:1000 | Rabbit | CellSignaling 5032S |
| DNMT3a | 1:1000 | Rabbit | ABclonal A19659 |
| DNMT3b | 1:1000 | Rabbit | ABclonal A11079 |
| NOS2 | 1:1000 | Rabbit | ABclonal A0312 |
| β-Actin | 1:1000 | Rabbit | CellSignaling 4970S |
| Anti-Mouse Secondary | 1:2000 | Horse | CellSignaling 7076S |
| Anti-Rabbit Secondary | 1:2000 | Goat | CellSignaling 7074S |

cleared mammary fat pad of 8-week-old female SCID Beige mice (Envigo). Only female mice were used as TNBC almost exclusively occurs in women. Upon reaching an average tumor volume of 150–250 mm3, the mice were randomly assigned to either the treatment or control group. For treatment, mice received intraperitoneal injections of L-NMMA (Santa Cruz Biotechnology) and amlodipine in sterile PBS (100 μL total/animal). L-NMMA was administered at 400 mg/kg on the first day and 200 mg/kg on subsequent days. Amlodipine (Major Pharmaceuticals NDC 0904-6371-61) was given at a dose of 10 mg/kg along with each L-NMMA treatment to counteract the elevated blood pressure associated with eNOS inhibition. The control group received sterile PBS via intraperitoneal injection (100 μL/animal).

**Western blot**

Whole cell lysates were obtained using the RIPA Lysis Buffer System (Santa Cruz), which includes lysis buffer, phenylmethylsulfonyl fluoride, protease inhibitor cocktail, and sodium orthovanadate. The protein concentration was determined using the Lowry assay (Biorad DC protein assay). Subsequently, 30–40 μg of lysate was loaded into each well of a 10-well 10% Mini-PROTEAN TGX precast gel with Laemmli sample buffer and β-mercaptoethanol. Electrophoresis was conducted at 115 V for 1 h. Protein transfer to a PVDF membrane was accomplished using the iBlot™ Transfer System (Invitrogen). The membrane was then blocked and incubated overnight at 4 °C with the primary antibody in 5% milk in PBS-Tween. Following secondary antibody incubation, each blot was imaged using the FluorChem E system (ProteinSimple). Chemiluminescent substrate coating, either Super-Signal™ West Femto Maximum Sensitivity Substrate (Thermo Scientific) or SuperSignal™ West Pico PLUS Chemiluminescent Substrate (Thermo Scientific), was applied before imaging. Densitometry was conducted using Fiji (ImageJ). The list of antibodies used is provided in Table 1.

**E. Coli expression/purification and in vitro TET2 enzymatic assays**

TET2 enzyme was expressed and purified from BL21 E. Coli (according to the protocol[49]) using a truncated 54.64 kD TET2 (1099-1936 with residues 1481-1843 replaced by a 15-residue GS linker). In vitro TET2 enzymatic assay was conducted by incubating 10 μM of an 8-nucleotide double-stranded DNA substrate (5′-CAC-X-GGTG-3′, where X is 5mC, 5hmC, or 5fC) with 5 μM of purified TET2 and varying concentrations of Sper/NO or DEA/NO. The reaction was allowed to proceed for 1–3 h at 37 C in a 25 μL assay also containing 50 mM HEPES

(pH 8.0), 100 mM NaCl, 100 μM Fe(NH4)2(SO4)2, 2 mM ascorbate, 1 mM DTT, 1 mM ATP, and 1 mM 2-KG. The DNA was desalted by adding 5 μL of AG® 50W-X8 Cation Exchange Resin (BioRad, Cat # 143-5441) directly into the biochemical mixture and agitated followed by incubation for 5 min at room temperature. The samples were then centrifuged at $9000 \times g$ for 2 min. 1 μL of sample was mixed with 1 μL of 3-Hydroxypicolinic Acid + Ammonium Citrate Dibasic matrix on a MALDI plate and the oxidized products were analyzed by MALDI-TOF mass spectrometry (Bruker-ultrafleXtreme™ MALDI-TOF/TOF spectrometer) using reflectron negative ionization mode and a mass range of 2400 - 2500.

**Dot blot for TET inhibition on genomic DNA**

For in vitro time-dependent enzymatic activity assay on genomic DNA, 10 μM wild type TET2 (1099–1936 del-insert) containing 50 mM HEPES (pH 8.0), 100 mM NaCl, 100 μM $Fe(NH_4)_2(SO_4)_2 \cdot 6H_2O$, 2 mM ascorbate, 1 mM DTT, 1 mM ATP, and 1 mM 2-OG was incubated with 0-300 μM Sper/NO (freshly prepared) at room temperature for 25 min. The demethylase assay was initiated with 1 μg of genomic DNA, isolated from HEK293T cells, and incubated at 37 °C for 3 h. After incubation, ¼ volume of 2 M NaOH–50 mM EDTA was added before addition of 1:1 ice cold 2 M ammonium acetate. Immobilin-P PVDF membranes were cut to size, wet with MeOH for 20 s, and equilibrated in TE buffer for 5 min, then assembled into a 96-well Bio-Dot microfiltration apparatus (Bio-Rad, Catalog #1706545). Each well was washed with 400 μL TE drawn through with gentle vacuum, and 400 ng of gDNA was loaded, followed by another TE wash. Membranes were blocked for 2 h in 5% milk–TBST washed 3× with TBST and blotted at 4 °C overnight with 1:3,000 anti-5hmC rabbit primary antibody (RRID-AB_10013602). Blots were then washed, incubated with secondary 1:5,000 goat anti-rabbit-HRP (Active Motif, cat. no. 15015) for 2 h. After another wash step, blots were imaged by chemiluminescence using VISIGLO HRP Chemiluminescent substrates A and B.

**ALKBH2 enzymatic assay**

To prepare lysates, nuclear and cytosolic fractions were extracted and combined using standard methods (according to the protocol[53]). 120 μL reactions were conducted in a 96-well plate at 37 °C in 50 mM HEPES buffer (pH 8.0) containing 75 μM $Fe(NH_4)_2(SO_4)_2 \cdot 6H_2O$, 1 mM α-ketoglutarate, 2 mM Sodium Ascorbate, 50 μg/ml BSA, 0.4 mg/ml Catalase (Bovine liver, Sigma), 2 μM probe (m1a demethylase-detecting fluorescent probe 13p[53]), and 0–500 μM SperNO (for recombinant protein reaction) or 0–200 μM DETA/NO (for whole cell lysate reaction)). The time course of fluorescence activation was recorded after addition of 500 nM AlkBH2 (either recombinant or purified cell lysate). Probe was excited at 355 nm and emission collected at 460 nm for 1 h for recombinant protein or 13 h for cell lysate.

**MALDI-TOF TET2 assay**

For in vitro enzymatic activity assays, TET2 (1099–1936 del-insert) containing 50 mM HEPES (pH 8.0), 100 mM NaCl, 100 μM $Fe(NH_4)_2(SO_4)_2 \cdot 6H_2O$, 2 mM ascorbate, 1 mM DTT, 1 mM ATP, and 1 mM 2-KG was incubated with 150 μM Sper/NO (freshly prepared), at room temperature for 25 min. The demethylase assay was initiated with 10 μM of double-stranded DNA substrate (5′-CAC-X-GGTG-3′, X-5mC, 5hmC, 5fC) and incubated at 37 °C for 3 h. Subsequently, the DNA was desalted by adding 8 μL of AG® 50W-X8 Cation Exchange Resin (BioRad, Cat. #143-5441) directly into the biochemical mixture and agitated followed by incubation for 5 min at room temperature. The samples were at $9000 \times g$ for 2 min. The oxidized products were analyzed by MALDI-TOF mass spectrometry (AB SCIEX Voyager DE Pro and Bruker-ultrafleXtreme™ MALDI-TOF/TOF spectrometer) by spotting 1 μL of sample and then mixed with 1 μL of 3-Hydroxypicolinic Acid (3-HPA) matrix on MALDI plate (with 3 technical replicates each).

Spectra were generated using reflectron negative ionization mode and a mass range of 2400-2500. Peaks were annotated using Bruker Daltonics flexAnalysis software with %5mC or %5hmC calculated as a proportion of the peak intensity of a certain adduct to the sum of all other adducts: 5mC (-2424.6), 5hmC (-2440.6), 5fC (-2438.6), 5caC (-2454.6). Experiments were completed with 3 biological replicates for each Sper/NO concentration.

To determine IC50 for Sper/NO against wt TET2, the assay mixture containing 5 μM protein was incubated with varying concentrations of Sper/NO (15 μM – 1,200 μM) in a buffer containing 50 mM HEPES (pH 8.0), 100 mM NaCl, 100 μM $Fe(NH_4)_2(SO_4)_2 \cdot 6H_2O$, 2 mM ascorbate, 1 mM DTT, 1 mM ATP and 100 μM 2OG for 20 min at room temp. Demethylation was initiated by adding 10 μM 8-nt double stranded 5mC DNA and further incubated at 37 °C for 2 h. The product DNA was processes as described above and by MALDI-TOF mass spectrometry. The values were fitted to the 4-S17 parameter non-linear regression algorithm (Y=Bottom + (Top − Bottom)/(1 + 10^((LogIC50 · X)*Hill slope))) of the GraphPad Prism software. X: log of dose or concentration; Y: response, decreasing as X increases; Top and bottom: upper and lower values of a given curve; logIC50: same log units as X; Hill Slope: Slope factor or Hill slope, unitless.

## 5mC and 5hmC ELISA

DNA was extracted from cells or tissues using the QIAGEN DNEasy Blood & Tissue kit and quantified using the BioTek Take3 system. Global %5mC and %5hmC were quantified by ELISA assay (Epigentek P-1030-96, P-1032-96). First, 100 ng DNA was bound to the bottom of each assay well. The wells were washed, and detection complex solution was added containing a 5mC or 5hmC antibody. After a 50 min incubation, wells were washed, and color developer solution was added before measuring absorbance at 450 nm using a BioTek Synergy HTX Multimode Reader. %5mC and %5hmC were calculated as a proportion of the total DNA.

## Electron paramagnetic resonance (EPR)

TET2 reactions with NO were conducted in a hypoxic chamber with 5% $CO_2$ and 95% $N_2$. Solutions were degassed with $N_2$ for one hour before bringing into the chamber. 28.8 μM purified truncated TET2 was incubated at 37 °C with 500 μM DEA/NO, 50 μM double-stranded methylated DNA substrate (5'-CAC-5mC-GGTG-3'), 100 μM ammonium iron(II) sulfate hexahydrate, 500 μM 2-oxoglutarate, 1 mM DTT, 2 mM ascorbate, 100 mM NaCl, 1 mM ATP, and 50 mM HEPES. Samples were flash frozen in liquid nitrogen either immediately after adding NO or 20 minutes after. X-band continuous wave EPR spectroscopy (a) at 77 K was conducted using a modified Varian E4 spectrometer equipped with a quartz finger dewar that operates at liquid nitrogen temperature, and (b) at 10 K using a Bruker ESP 300 spectrometer equipped with an Oxford Instruments ESR 910 continuous helium flow cryostat. Experimental parameters: for Fe(II)−NO signals 10 K, M.W. frequency 9.37 GHz, modulation amplitude 5 G; for DNIC spectra T = 77 K, M.W. frequency 9.135 GHz, modulation amplitude 10 G.

## RNA sequencing (RNA-seq) and analysis

Total RNA was isolated using the RNAqueous™-4PCR kit (Ambion) and treated with DNase I (Ambion) to avoid genomic DNA contaminations. Libraries were prepared with Kapa Hyper Stranded mRNA library kit (Roche) with adaptor sequence AGATCGGAAGAGCACACGTCTGAA CTCCAGTCACNNNNNNNNNATCTCGTATGCCGTCTTCTGCTTG (NNNN NNNN = 8 nt index). The libraries were pooled; quantitated by qPCR and sequenced on one SP lane for 101 cycles from one end of the fragments on a NovaSeq 6000. Fastq files were generated and demultiplexed with the bcl2fastq v2.20 Conversion Software (Illumina). Library preparation and sequencing was conducted by the University of Illinois at Urbana Champaign Roy J. Carver Biotechnology Center in triplicate, with a total of 476 million reads 100 nt in length.

Read quality was assessed using MultiQC. Reads were aligned to hg38 GENCODE human genome (release 22) using STAR 2.5.2a[110]. Bam files were sorted using samtools[111]. Gene counts were generated using HTSeq[112]. Differential expression analysis was completed using quasi-likelihood F-tests (glmQLFTest function) in edgeR v3.40.2[113]. Gene Set Enrichment Analysis was completed using OmicPath (https://github.com/CBIIT-CGBB/OmicPath).

## Oxidative reduced representation bisulfite sequencing (OxRRBS) and analysis

Genomic DNA was extracted and purified from cells using the DNEasy Blood & Tissue kit (QIAGEN) with RNase step. The libraries were prepared with the Ovation® RRBS Methyl-Seq with TrueMethyl® oxBS from Tecan with adaptor sequence AGATCGGAAGAGC. The libraries were pooled; quantitated by qPCR and sequenced on one S1 lane for 101 cycles from one end of the fragments on a NovaSeq 6000. Fastq files were generated and demultiplexed with the bcl2fastq v2.20 Conversion Software (Illumina). Oxidative reduced representation bisulfite sequencing library preparation and sequencing was performed by the UIUC core in duplicate with a total of 924 million reads 20–100 nt in length. Read quality was assessed using MultiQC[114]. Sequences were trimmed using TrimGalore (https://github.com/FelixKrueger/TrimGalore). Reads were aligned to hg38 using bowtie2 within Bismark (mapping efficiency ~62%) to count 5mC and 5hmC marks in a CpG context. Methylation analysis was performed using Bismark[115] and Rnbeads[116]. Differential methylation analysis was performed using RnBeads, Dnmtools[117], and MethylKit[118]. CpG annotation was performed with the annotatr package[119] from BioConductor.

## Density functional theory simulations and modeling

Density functional theory simulations employed the Gaussian 16 code[120] to investigate the stability of nitrosyl complexes of Fe(II) active site models of TET2. As per the study of Lu et al.[57], the ωB97xD functional was employed as this Hamiltonian incorporates dispersion effects[121], in conjunction with the def2-svp basis set[122] for geometry optimizations and vibrational frequency calculation. For more accurate reaction energies, a single point calculation with a larger basis set – def2-tzvpp – was employed at the optimized stationary points. For the model complexes investigated all reasonable charge, spin and coordination states were examined. The results presented herein focus on the ground states identified for the proposed reaction intermediates. All complexes were fully optimized in the absence of any geometric or symmetry constraints. Quoted thermodynamic values assumed a temperature of 298.15 K and a pressure of 1 atm; enthalpic and entropic corrections used unscaled vibrational frequencies obtained at the ωB97xD/def2-svp level of theory. An SMD continuum solvation model (solvent = water) was employed for closer congruence with experimental conditions[122].

The DNIC was modeled into PDB file 4NM6[58] using the modeling program COOT[123]. The model was energy minimized using REFMAC[124] as implemented in CCP4i[125]. The NO bond length was restrained to 1.12 Å and the Fe-NO bond lengths restrained to 2.0 Å.

## Modeling simulations of NO, $O_2$, and NO donor concentrations

To simulate the kinetics of the NO donor decomposition and steady-state NO concentrations we utilized differential equation-solving that involved two main processes: the first-order decay of the NO donor molecule (1.5 equivalents of NO/DEA/NO $t_{1/2}$ = 16 min, 2 equivalents of NO/Sper/NO $t_{1/2}$ = 230 min, 2 equivalents of NO DETA/NO $t_{1/2}$ = 57 h)[126] and the autooxidation of NO in the presence of $O_2$ to form nitrogen dioxide ($NO_2$). The decay constant k for the NO donors were calculated using the formula: k NO donor = ln(2) / half-life. Autooxidation of NO: The reaction between NO and $O_2$ to form $NO_2$ follows third-order kinetics with the rate law: Rate = $k[NO]^2 [O_2]$ where k = 2 ×$10^6$ $M^{-2}s^1$. Initial concentrations were set as follows: [NO donor]$_0$ = starting NO

donor concentration for each simulation, $[O_2]_0 = 220\,\mu M$, $[NO_2]_0 = 0\,\mu M$. The steady-state NO concentration was then determined by calculating the difference in the rate of NO liberation from the donor and the rate of NO consumption (assumed to be predominantly autooxidation in aqueous solution). Given by Equation (1):

$$\frac{d[NO]}{dt} = k_1[D]e_{NO} - k_2[O_2][NO]^2$$

Where [NO] is the concentration of NO at given time $t$, $k_1$ is the rate constant for the decomposition of the donor, $[D]$ is the concentration of the donor at time $t$, $e_{NO}$ is the factor representing moles of NO release per mole of donor, $[O_2]$ is the oxygen concentration, and $k_2$ is the rate constant for oxidation of NO by oxygen (autooxidation)[127]. The system of differential equations was solved numerically using the NDSolve function in the Wolfram Language. Solutions for the concentrations of NO donors, NO, and $O_2$ were evaluated at regular intervals (seconds). The model does not assume a constant $O_2$ concentration. Instead, it includes a differential equation for the consumption of $O_2$ in the autooxidation reaction. These equations collectively account for the dynamic changes in the concentrations of NO donors (DEA/NO, Sper/NO, DETA/NO), NO, and $O_2$ over time.

### Reporting summary

Further information on research design is available in the Nature Portfolio Reporting Summary linked to this article.

## Data availability

All relevant data supporting the key findings of this study are available within the article and its Supplementary Information files. Minimum datasets are provided that are necessary to interpret, verify, and extend the research in the article are provided. Source data are provided with this paper. RNA-sequencing and RRBS-sequencing data are stored in the NCBI GEO repository under accession number: https://www.ncbi.nlm.nih.gov/geo/query/acc.cgi?acc=GSE248151 Source data are provided with this paper.

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

## Acknowledgements

Research reported in this publication was supported by in part by: the National Institute of General Medical Sciences of the National Institutes of Health under award number R01GM155946 (D.T.). University of Illinois Chicago Department of Pharmaceutical Sciences Bridge Fund, Hans W. Vahlteich Bridge Fund (D.T.), University of Illinois at Chicago Drug Discovery Center (UICentre), and the University of Illinois Cancer Center (D.T.). National Institutes of Health (R01GM111097) (B.M.H.). This work utilized the computational resources of the NIH HPC Biowulf cluster (http://hpc.nih.gov) (M.S.). National Cancer Institute grant CA217809 (E.T.K.). This work was supported, in whole or in part, by NCI grant no. U01 CA268813 and R01 CA284315, the Breast Cancer Research Foundation (BCRF), CREDO, and philanthropic support from Dr. M. Neal and R. Neal (J.C.). The Intramural Research Program of the NIH, NCI, Center for Cancer Research and Cancer Innovation Laboratory (D.W.M.).The National Science Foundation (CHE-2204114) and the National Institutes of Health (R01GM130752) (K.I.). DFT modeling used the NSF-funded (OAC-2117247) HPC resource at UNT (T.C.). Science Foundation Ireland grant number 17/CDA/4638 (S.G.). This work was supported by the National Institutes of Health grant U54 CA143924 Sub-Project 676 (W.M.).

## Author contributions

D.T. conceived and supervised the project. M.P., H.K., S.S., K.I., performed the in vitro and in vivo experiments. M.P., B.H., D. McVicar, and D.T. participated in experimental design and data analysis. T.C. and W.M. performed the DFT simulations and modeling. D.W. and E.K. provided fluorescent probes and methodology. C.C. and A.P. helped with study design and data processing. B.H., V.K., and H.Y. performed the EPR analysis. S.G. provided the NOS2 cells and technical advice. J.C. provided tumor samples. Y.F., Q.C., D. Meerzaman, and M.S. conducted the bioinformatic and sequencing analysis. D.T. and M.P. wrote the manuscript with input from all coauthors.

## Competing interests

J.C. is the sole inventor on patent application no. 10420838 entitled: Methods for treating cancer using iNOS-inhibitory compositions, held by Houston Methodist Hospital. The other authors declare no competing interests.
