## [Peer Review file · Nature Communications]

Nitric oxide inhibits ten-eleven translocation DNA demethylases to regulate 5mC and 5hmC across the genome

Corresponding Author: Professor Douglas Thomas

Version 0:

Reviewer comments:

Reviewer #1

(Remarks to the Author)

This manuscript reports that nitric oxide can inhibit TET dioxygenases and ALKBH2 repair protein by binding to non-heme iron and preventing cofactor binding. This is a potentially exciting finding that would directly link inflammation with epigenetic deregulation. Unfortunately, the cellular data are not consistent with the conclusion because NO treatment induced an increase in both MeC and hmC (e.g. Fig 3C). TET inhibition should decrease hmC levels as reported for many other inhibitors. The authors provide no explanation for this contradictory result not consistent with their hypothesis. The inhibition data for in vitro experiments appears fragmented/incomplete, and some of the figures are crowded and are not explained well. I believe that publication is premature at this point.

Reviewer #2

(Remarks to the Author)

Palczewski and colleagues investigated the effect of NO on DNA methylation and demonstrate that NO inhibits DNA methylases (TET and ALKBH2) to increase levels of 5mC and 5hmC in DNA. They characterize this inhibition of TET as reversible and concentration dependent and due to the formation of a dinitrosyl iron complex at the enzyme's active site. They then show that TET inhibition leads to increased DNA methylation at gene-regulatory loci in cancer cells, which was associated with the differential expression of NO-regulated cancer genes. The study is interesting, well executed and describes a novel mechanism of NO signaling.

1- The data with different NO donors/timing in Figure 1D is interesting and consistent with reversibility. An experiment in which NO is intentionally removed (via a scavenger) would more directly test reversibility.

2- Quantification of Figure 1E would be helpful for the reader.

3- Figure 1 legends states that data are means. Error bars for panel G would be helpful.

4- In Figure 3, Panel E shows a doubling of NO concentration from 24 to 48 hours, yet Panel F shows no significant change between 5mC over the same time period. Is there an explanation for this disconnect?

Reviewer #3

(Remarks to the Author)

Can the authors justify that the concentration of NO donors studied are relevant to health or disease. For example, do the NOS2 high triple negative breast cells generate comparable concentrations? Is it possible to use a drug or neurohormonal stimulation that stimulates NOS activity and detect changes in enzyme modification, activity, and cytosine methylation status? Perhaps overexpression of NOS can be used to strengthen the concept that NO generated by cells is sufficient to generate the phenotype via the mechanism proposed. I realise that NOS2 high breast cells were studied, but they will have many other alterations that may contribute to their aggressive phenotype.

What is the expected concentration half constant for NO to form the dinitrosyl iron complex? How does that compare with the concentration-dependent response on enzyme activity? Is there any way to ascertain the stoichiometry of the iron

nitrosylation? My understanding is the iron in soluble guanylyl cyclase has very high affinity for NO and would be activated by much lower concentration of NO that is needed to change enzyme activity here. So, I am wondering if physiologic NO can indeed target the enzymes studied here, or is it simply only relevant when iNOS is expressed during inflammatory disease scenarios?

The aggressive cancer phenotype induced by chronic NO exposure correlates with alteration in methylation changes at specific loci. There will be many other signaling events and changes in these cells during this treatment, and so I am wondering if anything can be done to connect these together to strengthen causality. For example, expression changes have been detected - can some of these proteins that may rationally explain the aggressive phenotype be targeted to attenuate the transition? For example, a protein or pathway that is upregulated and may explain the phenotype, might be genetically or pharmacologically targeted and it could be determined if this attenuates the aggressive cancer phenotype?

Reviewer #4

(Remarks to the Author)

The manuscript by Palczewski and coworkers describes the interaction of NO with ten-eleven translocation DNA demethylases. The data extend logically from purified samples to cultured cells to significant in vivo analysis. While the findings are impactful, the manuscript itself is sometimes too cursory to the point of being confusing. A key example is a lack of discussion of the NO donors used, including the relationship to NO concentrations. Another is the use of a neutral complex in modeling, presumably by adding a random electron. There is also a surprising dearth of references in the introduction. The figures are also quite small. In general, however, the experiments are logical, the data are informative and the conclusions are significant with regard to the impact of NO on demethylation, especially in cancer cells. Specific comments follow.

The chemistry of NO is rather cursorily described in the introduction, including the fact that NO does not directly induce nitrosation.

Results should be removed from the introduction.

In Figure 1A, inclusion of stoichiometric analysis would be beneficial.

Too many significant figures are reported, beginning with Figure 1B.

What is the error in Figure 1C?

Were controls performed with expired SPER/NO?

In Figure 1D, does the negative sign indicate that DEA/NO was present for 3 hours, just expired during this time period? Again, stoichiometric analysis would be beneficial for this data set. Is there a reason that 50% inhibition is observed for both donors at 1 h? Is a longer time period for the SPER/NO experiment feasible?

Starting at line 162, it would be beneficial to describe the bonding site. That being said, binding of iron by NO is well-known and thus does not have to be rationalized.

Figure 2A: a spectrum for the enzyme alone should be included.

Starting at line 184, more details need to be provided here including assignment as a triplet.

Starting at line 229, epigenetic inheritance is mentioned. Yet, NO leads to an increase in methylated product while the level is unchanged in the absence of NO. This suggests susceptible vs. non-susceptible demethylations sites. Analysis with NO production inhibited could be informative.

Figure 3E would again benefit from stoichiometric analysis or at least discussion relative to physiological levels.

Figure 3G would benefit from some work with ChemDraw.

In the experiment starting on line 296, NO levels should be described.

In the discussion of previous studies, are there any clear differences, such as NO concentration, tissue type, etc?

Line 390 – with NO bound perhaps would be better.

In response to the final discussion, it would be informative, although not necessary to this study, to compare to nontumorigenic cell lines such as MCF-10.

Version 1:

Reviewer comments:

Reviewer #1

(Remarks to the Author)

The authors provided a revised version of the manuscript, however they failed to address the main concern that, according to their data, inhibition of TET by NO leads to increased levels of hmC in cells. TET inhibition should reduce the levels of hmC in genomic DNA, which calls the paper's conclusions into question. One possible explanation for this puzzling result is the limitations of the methodology used in this paper. They claim that they used a "rigorous approach" to quantify hmC, however, ELISA can hardly be considered a quantitative method for hmC, and no independent confirmation of this results was provided. A golden standard in the field is isotope dilution HPLC-MS/MS to provide absolute accurate and precise levels of hmC and other epigenetic marks in cells, but this was not done in this study.

The authors acknowledge that their "long-term" NO exposure experiments using the O donor DETA/NO repeatedly added it to the cells every 48h led to wide fluctuations in NO levels and could result in artifactual results. It is not clear why NO gas could not be used directly for these experiments.

The argument about hypoxia is hardly relevant here as the cells were not grown under hypoxic conditions. However, the point about the need to measure absolute rather than relative levels of meC and hmC is well taken and further emphasizes the requirement for accurate measurements using mass spectrometry to confirm the results obtained from antibody based methods such as ELISA. Such methodologies are widely available and have been used by many laboratories, including studies investigating TET inhibition by small molecules (Guan et al, Blood Cancer Discov. 2021 Mar;2(2):146-161). As expected, these authors observed a reduction in hmC levels in cells treated with TET inhibitors.

Reviewer #2

(Remarks to the Author)

The reviewers have adequately addressed all my concerns and the manuscript is now strong.

Reviewer #3

(Remarks to the Author)

The authors have sufficiently addressed my comments with replies to my questions that were clear and rational.

Reviewer #4

(Remarks to the Author)

The authors responded in significant detail, including with inclusion of additional data, to the many points of the reviewers. As such, the manuscript has substantially improved in terms of soundness, experimental details and legibility, clarity to the reader, demonstration of significance, etc. As such, the recommendation is to publish at this point.

Version 2:

Reviewer comments:

Reviewer #1

(Remarks to the Author)

The authors have addressed my previous concerns by providing LC-MS data for hmC in cells treated with nitric oxide. The revised manuscript is suitable for publication.

RESPONSES TO THE REVIEWERS' COMMENTS

Reviewer #1 (Remarks to the Author):

1. *This manuscript reports that nitric oxide can inhibit TET dioxygenases and ALKBH2 repair protein by binding to non-heme iron and preventing cofactor binding. This is a potentially exciting finding that would directly link inflammation with epigenetic deregulation.*

We appreciate the reviewer's recognition of the potential significance of our findings.

2. *Unfortunately, the cellular data are not consistent with the conclusion because NO treatment induced an increase in both MeC and hmC (e.g. Fig 3C). TET inhibition should decrease hmC levels as reported for many other inhibitors. The authors provide no explanation for this contradictory result not consistent with their hypothesis.*

We acknowledge the reviewer's concern regarding the observed increase in both 5mC and 5hmC levels upon NO treatment. As the reviewer pointed out, there are many examples in the literature demonstrating that TET inhibition results in decreased levels of 5hmC. Our data demonstrated robust increases in 5mC and 5hmC upon cellular exposure to NO (exogeneous **Fig. 3A,B** or endogenous **Fig. 3F**) that was consistent across 4 different cell types (**Fig. 3A**), at 2 time points (24h and 10 days, **Fig. 3A-C**), and as measured by two separate methodologies (ELISA, and 5mC- 5hmC-Sequencing). Because of this rigorous approach and the consistent findings, we have confidence in our results, however, we strongly agree that it is important to discuss differences in our data in relation to findings by other groups and to provide plausible mechanisms to explain our findings:

A recent paper by Belle et. al. in *J. Med. Chem* 2024¹ examined the sensitivities of human TET enzymes to the inhibitor and oncometabolite 2-Hydroxyglutarate. They found that “*Most inhibitors manifested similar potencies for TET1–3 and caused increases in cellular 5hmC levels*”. They also noted that different TET isoforms exhibited different sensitivities to inhibition, in isolated enzymes and in cells. A similar effect was observed by Putiri et. al (*Genome Biol* 2014)² in an embryonic carcinoma cell model where TET1 depletion yielded global reduction of 5hmC while TET2,3 depletion caused increased 5hmC. Differences in TET isoform-specific inhibition by NO was not tested in our studies, but given that all 3 TET enzymes were expressed in our cell models, this could potentially contribute to increases in 5hmC under specific circumstances. Our studies were the first to measure the effects of long term (10 day) NO treatment on epigenetic processes (and on transcriptional changes). Although we were meticulous about exposing the cells to physiological steady-state NO concentrations, these treatment paradigms are methodologically challenging. To achieve “long-term” NO exposure we used the NO-donor DETA/NO ($t_{1/2} \sim 22$ h) and repeatedly added it to the cells every 48 h. Based on the release kinetics of DETA/NO (**see new Extended Data Figure 1D**), the cells are, in reality, exposed to a range of NO concentrations that decreases over 48 hours until new donor is added (we have extensively characterized NO-donor release kinetics *in vitro* and *in vivo*³⁻⁵). Therefore, it is possible that over the course of 48 h, NO levels may fall below the minimum inhibitory concentration for one or all the TET enzymes and allow for some 5mC→5hmC conversion. In general, this is interesting and mimics *in vivo* conditions where NO synthesis can be turned on or off either at the level of expression (NOS2) or its activation (eNOS, Ca²⁺/calmodulin).

Another potential explanation for NO-mediated increases in 5hmC can be learned from studies on hypoxia. Oxygen (O₂) is a substrate for TET enzymes, and therefore TET enzymes are known to be catalytically inactive in physiologically hypoxic environments (below the K_M for O₂). Despite this fact, numerous reports using different model systems have demonstrated that hypoxia increases 5hmC levels in cells, which, similar to our data, seems counterintuitive. One of the first reports of this phenomenon was by Mariana et. al. in *Cell Rep* 2014⁶, where they demonstrated that “*hypoxia increases global 5-hmC levels, with accumulation of 5-hmC density at canonical hypoxia response genes.*” He et. al. (*Oncol Lett.* 2021⁷) similarly demonstrated using human acute myeloid leukemia cells cultured at 3% and 1% O₂ that there was a dose-dependent increase in 5hmC and a concomitant decrease in 5mC at 24, 48, and 72h. Prasad et. al. (*Stem Cells.* 2017⁸), demonstrated that in glioma cells, hypoxia (0.2% O₂ at 24, 48, 72 h) induced TET1/TET3 expression and 5hmC clusters at gene regulatory regions. In HepG2 liver cancer cells, Lin et. al. (*Oncol Lett.* 2017⁹) showed that hypoxia (1% O₂) induced the expression of TET enzymes and increased 5hmC. Another study (Wu et. al., *Cancer Research* 2015¹⁰) revealed that hypoxia increased global 5hmC levels in MDA-MB-231 cells (one of the same cell lines in our study) as well as in tumors. Potentially important insight into our results comes from Cao et. al. (*Blood Advances* 2020¹¹) where they cultured cells for 10 days under hypoxia. Intriguingly they found that global levels

of 5hmC were decreasing at both 1% and 21% oxygen over 10 days, but that relative levels of 5hmC were significantly enriched at gene regulatory loci at 1% O₂ compared to 21% O₂. In our study, we compared “relative” levels of 5mC/5hmC in NO-treated cells to cells not treated with NO at each time point. Therefore, it is possible that absolute levels of 5mC/5hmC could have decreased over time, but relative levels were increasing.

One mechanism by which NO exerts its biological activity is by replacing O₂ at the oxygen-binding sites of heme and non-heme proteins (such as TET) to inhibit their catalytic activity. Therefore, adding NO is like taking away O₂ (hypoxia) (reviewed by us¹²). For this reason, NO is sometimes referred to as a “hypoxia mimetic”. Although the term is not accurate in a technical sense, NO can bind to and inhibit the hypoxia-inducible factor (HIF) prolyl hydroxylase (PH) which results in HIF1 α accumulation and the induction of down-stream hypoxic responses. We^{13,14}, and others¹⁵, have demonstrated that under normoxic conditions (21% O₂) NO is a potent inhibitor of PH leading to HIF-1 α accumulation. McCarthy et. al. (*PLoS One* 2015¹⁶) treated AML cell lines with cobalt chloride (CoCl₂), which is another chemical “hypoxia mimetic” that binds to non-heme iron enzymes in a manner similar to NO. They reported that at 21% O₂, CoCl₂ significantly increased 5hmC. One explanation, cited by many of the papers for increased 5hmC under hypoxia, was that HIF-1 α increases the expression levels of some TET isoforms. However, we showed no change in the expression of TET1,2,3 in response to NO, which is consistent with other studies demonstrating that expression profiles of the TETs do not in fact mirror the dynamic changes in global 5hmC levels^{17,18}.

Considering other explanations outside of hypoxia, additional data in the manuscript by Belle et. al.¹ showed that the HDAC inhibitor Panobinostat increased 5hmC levels in cells in a concentration- and TET1-activity-dependent manner. They theorized that HDAC inhibition, which results in elevated levels of histone lysine acetylation, may improve the accessibility of DNA to TET enzymes as well as increase TET expression. This is potentially relevant to our data, as we have demonstrated that physiologic NO concentrations can regulate histone posttranslational modifications in cells including increasing acetylation at specific lysine residues (i.e., H3K27)¹⁹⁻²¹.

Another possibility is that NO reduces the chemical processivity of TET, that is, the ability of a TET enzyme to convert an individual CpG from 5mC to 5caC without releasing its substrate. Various studies have revealed that the chemical processivity of TET may depend on the availability of cofactors, interactions with partner proteins, chromatin accessibility, and other factors. Though a significant portion of TET will oxidize only 5mC and then release it to produce 5hmC, a portion will stay attached to the substrate and convert 5mC to 5caC. However, if NO is decreasing the chemical processivity of the enzymes and causing TET to release the DNA before fully converting to 5caC, a conceivable increase in 5hmC or 5fC may occur as long as TET chemical processivity remains inhibited.

Other potential mechanisms to explain NO-mediated increases in 5hmC are: changes in the availability of essential cofactors like 2-OG, 5mC might be a better substrate for TET than 5hmC, reduction of competitive endogenous inhibitors like 2-HG, genetic modifications, and specific microenvironmental conditions that regulate the activity of TET enzymes. Although we favor a direct mechanism of TET inhibition by NO to explain the majority of increases in 5hmC, we cannot fully rule out other concurrent mechanisms. We have added the following new text to the discussion section to highlight both the differences and similarities between our results and published literature. We hope this puts our data in better context and provides the readership with plausible mechanisms for the observed increases in 5hmC in response to NO.

“Our cellular studies demonstrated that NO caused global, and loci-specific, increases in both 5mC and 5hmC in DNA. It is expected that 5mC would increase consequent to TET inhibition, yet mechanisms to explain why 5hmC also increases are less intuitive. Insight to this comes from a recent paper by Belle et. al.⁷⁹ where they examined the sensitivities of human TET enzymes to various inhibitors. They found that most inhibitors manifested similar potencies for all TET isoforms and caused an increase in 5hmC in isolated enzymes and in cells. Another group found similar results using an embryonic carcinoma cell model where TET2,3 depletion caused increased 5hmC⁸⁰. An alternate explanation for NO-mediated increases in 5hmC can be learned from studies on hypoxia. Oxygen (O₂) is a substrate for TET enzymes, and therefore TET enzymes are known to be catalytically inactive in physiologically hypoxic environments (below the K_M for O₂). Despite this fact, numerous reports have demonstrated that hypoxia increases 5hmC levels in cells, which, similar to our data, seems counterintuitive. One of the first reports of this phenomenon in 2014 demonstrated that hypoxia increased global 5hmC in DNA that accumulated at canonical hypoxia response genes⁸¹. Since then, numerous studies have replicated these findings in various model systems and demonstrated that both short (24 h) and long-term (10 day) hypoxia increases 5hmC in DNA⁸²⁻⁸⁶. One explanation cited for increased 5hmC under hypoxia was that hypoxia increased

the expression levels of some TET isoforms. We explored this possibility and determined that concentrations of NO that increased 5hmC did not change the expression levels of TET1,2, or 3. This is consistent with other studies demonstrating that expression profiles of the TETs do not mirror dynamic changes in global 5hmC levels^{87,88}. In another study, researchers treated cells with cobalt chloride (CoCl₂), a chemical “hypoxia mimetic” that targets many of the same metalloproteins as NO. Consistent with our results, they measured a significant increase in 5hmC under normoxia (21% O₂)⁸⁹.

Considering other explanations for increases in 5hmC, it has been shown that the HDAC inhibitor Panobinostat increased 5hmC levels in cells in a concentration- and TET1-activity-dependent manner⁷⁹. They theorized that HDAC inhibition, which results in elevated levels of histone lysine acetylation, may improve the accessibility of DNA to TET enzymes as well as increase TET expression. This is potentially relevant to our data as we have demonstrated that physiologic NO concentrations can regulate histone posttranslational modifications in TNBC cells, including increasing acetylation at specific lysine residues (i.e., H3K27)^{42,43,61}. Although we favor a direct mechanism of TET inhibition by NO to explain the majority of increases in 5hmC, we cannot fully rule out other concurrent mechanisms.”

3. *The inhibition data for in vitro experiments appears fragmented/incomplete, and some of the figures are crowded and are not explained well. I believe that publication is premature at this point.*

We appreciate the feedback regarding the inhibition data and figures. We have revised and enlarged the figures to improve clarity, ensure completeness, and provide a more detailed explanation in the figure legends and text. Specifically in regard to the *in vitro* inhibition data, we have completely updated **Figure 1** and included new data and the following text:

“Because of the small reaction volumes used for these experiments, we were not able to directly measure steady-state NO concentrations ([NO]_{ss}). NO-donor compounds, such as Sper/NO, are very useful for studying biological effects of NO because of their predictable release kinetics. However, the steady-state NO concentration cannot be accurately extrapolated from the starting concentration of the NO-donor. To estimate the inhibitory [NO]_{ss} for TET, we solved a system of differential equations that collectively considered dynamic changes in the concentrations of NO-donors, NO, and O₂ over time. In an aqueous solution with isolate enzyme, the [NO]_{ss} is determined by the difference in the rate of NO liberation from the donor and the rate of NO consumption (predominantly autooxidation). Based on these simulations, the IC₅₀ for [NO]_{ss} was approximately 3 μM for isolated TET2 enzyme under our assay conditions (**Extended Fig. 1A**). Having established that NO could inhibit the first step of TET’s three sequential oxidations (5mC ⇒ 5hmC) we tested whether NO could also inhibit the stepwise oxidation of 5hmC ⇒ 5fC, and in a separate reaction, the conversion of 5fC ⇒ 5caC. We conducted two separate reactions, one that used 5hmC as the starting TET2 substrate and the other that used 5fC as the initial substrate. Both reactions were treated with the IC₅₀ concentration of Sper/NO for TET2 (165 μM). When the reaction was started with 5hmC-DNA, the enzymatic conversion to 5fC was inhibited by 23%. When the reaction was initiated with 5fC-DNA as the TET2 substrate, conversion to 5caC was inhibited by 20% (**Fig. 1C**). We next asked whether inhibition of TET2 by NO was permanent or reversible. To address this question, we measured TET2 activity (conversion of 5mC to 5hmC) over a 3-hour period under three different treatment conditions: **1**) TET2 in the absence of NO, **2**) TET2 incubated with NO for the entire 3 h experiment, and **3**) TET2 incubated with NO for less than 3 hours. The concentrations of NO and the durations of NO exposure were established using two different NO-donor compounds; DEA/NO, a short-acting NO donor (t_½ ≈ 16 min at 25° C), and Sper/NO, a longer-acting NO donor (t_½ ≈ 230 min at 25° C)⁵⁰. Mathematical modeling was used to simulate theoretical steady-state NO concentrations from each NO-donor under each experimental reaction condition (bottom graph panels in **Fig. 1D**). The simulated kinetics and steady-state NO concentrations were similar to what we previously determined experimentally using the same NO-donors^{15,50-52}. Predicted changes in the concentrations of O₂ and NO-donor compounds were also modeled (**Extended Data Figure 1B, C**). When we measured TET2 demethylase activity in the absence of NO, TET2 was fully active and resulted in 100% conversion of 5mC ⇒ 5hmC within the first hour (“Full activity” **Fig. 1D.1**). We then exposed TET2 to a continuous [NO]_{ss} for the entire 3 h duration of the experiment, and its demethylase activity was 100% suppressed throughout the experimental timeframe (“Full Inhibition” **Fig. 1D.2**). Next, we incubated TET2 with the NO-donor DEA/NO, which, due to its rapid release kinetics, produced an initial burst of NO that disappeared by the end of the 3-hour experiment. Under this condition, TET2 demethylase activity was

inhibited completely at 1 hour when the NO concentration was high. However, by 2 hours, when the NO concentration had decreased, we observed recovery of enzyme activity (~50% product formation), and at 3 hours the reaction was 100% complete (“Inhibition and Recovery” **Fig. 1D.3**). Since TET2 enzymatic activity was restored between 1 and 2 hours, this suggested that the “minimum inhibitory concentration” for TET2 was within the range of 0.6 - 1.65 μ M NO. The observed recovery of enzyme activity following NO disappearance suggested that the inhibitory effect is reversible and not due to permanent covalent modifications or degradation of the enzyme. Lastly, instead of using a synthetic 5mC-DNA oligo substrate, we tested NO-mediated TET2 inhibition using its biological substrate: genomic DNA (**Fig. 1E**). NO inhibited TET2-catalyzed conversion of 5mC to 5hmC in genomic DNA at similar Sper/NO concentrations as when 5mC-DNA oligos were used as substrates.”

Reviewer #2 (Remarks to the Author):

4. *Palczewski and colleagues investigated the effect of NO on DNA methylation and demonstrate that NO inhibits DNA methylases (TET and ALKBH2) to increase levels of 5mC and 5hmC in DNA. They characterize this inhibition of TET as reversible and concentration dependent and due to the formation of a dinitrosyl iron complex at the enzyme’s active site. They then show that TET inhibition leads to increased DNA methylation at gene-regulatory loci in cancer cells, which was associated with the differential expression of NO-regulated cancer genes. The study is interesting, well executed and describes a novel mechanism of NO signaling.*

We appreciate the reviewer's supportive feedback regarding the potentially important implications of our findings.

5. *The data with different NO donors/timing in Figure 1D is interesting and consistent with reversibility. An experiment in which NO is intentionally removed (via a scavenger) would more directly test reversibility.*

The reviewer makes an excellent suggestion. In regard to **Figure 1D**, please see response #3 to Reviewer 1. We have now improved the presentation of this data and included new data for **Figure 1D** and supporting data in **Extended Figure 1**. We also clarified these results in the text.

6. *Quantification of Figure 1E would be helpful for the reader.*

Thanks for this suggestion, the data in **Figure 1E** have been quantified and the figure updated.

7. *Figure 1 legends states that data are means. Error bars for panel G would be helpful.*

We have included error bars for panel G as suggested (as well as panel C).

8. *In Figure 3, Panel E shows a doubling of NO concentration from 24 to 48 hours, yet Panel F shows no significant change between 5mC over the same time period. Is there an explanation for this disconnect?*

Panel 3E is measuring nitrite (NO₂⁻) which is a stable NO oxidation product that accumulates in the media of cells that synthesize NO. A doubling of nitrite over 24-48 h (which we would expect) suggests that the cells are continuously synthesizing NO at a steady rate. Unfortunately, we cannot directly infer the NO concentration from the nitrite concentration, as the nitrite concentration only tells us how much total NO has been synthesized over a period of time. We suspect that the maximum inhibitory concentration of NO for TET was achieved within the first 24 hours and therefore we would not expect to see any further increases in 5mC from 24-48 hours.

Reviewer #3 (Remarks to the Author):

9. *Can the authors justify that the concentration of NO donors studied are relevant to health or disease. For example, do the NOS2 high triple negative breast cells generate comparable concentrations?*

We appreciate this question as we have dedicated a major portion of our research over the past decades studying concentration-dependent effects of NO^{3-5,13}. We use NO-donor concentrations that generate steady-state NO concentrations within the physiological/pathological range. As a general rule, we consider this to be in the low micromolar to nanomolar range. For example, in **Figure 3** we used 100 μ M of the NO-donor DETA/NO. We know from our previous studies (and modeling **Extended Data Figure 1**), where we measured [NO]_{ss} over

time by electrochemical and chemiluminescence detection, that this NO-donor concentration corresponds to approximately 50-700 nM NO.

Regarding the NOS2-expressing cells that we used, for clarification, we do not consider these to be “NOS2 high” TNBC cells. These cells simply express some NOS2 protein compared to the wild type cells that do not express any NOS2 protein. When we originally characterized these cells, we found that over a 48 h period, they actually generated slightly less NO than was liberated from 100 μ M of the NO-donor DETA/NO over 48 h. As mentioned above, total NO synthesis over time can only give us an estimate of the actual steady-state NO concentrations the cells are exposed to. Nevertheless, we feel confident that when we are exposing cells to NO, the cells are experiencing physiologically relevant NO concentrations from either NO-donors or from endogenous synthesis when NOS2 expressing cells are used.

10. *Is it possible to use a drug or neurohormonal stimulation that stimulates NOS activity and detect changes in enzyme modification, activity, and cytosine methylation status? Perhaps overexpression of NOS can be used to strengthen the concept that NO generated by cells is sufficient to generate the phenotype via the mechanism proposed. I realize that NOS2 high breast cells were studied, but they will have many other alterations that may contribute to their aggressive phenotype.*

Although TNBC tumors from human patients express NOS2, most of the immortalized TNBC cell lines used for cell culture experiments do not. To address this discrepancy, we previously devised a method to stimulate TNBC cells (MDA-MB-231) in culture to upregulate NOS2 and synthesize NO. This technique was cumbersome and required administering cytokines to the cells for extended periods of time. Although this technique had some advantages, as the reviewer has pointed out, we found that the cytokines themselves induced too many gene expression and phenotypic effects that we could not attribute solely to NO synthesis. For these reasons, to precisely control the concentration and duration of NO exposure we prefer to use NO-donors, and then validate key findings in the NOS2-transfected cells that do not require additional stimulation for NO synthesis. Moreover, the mouse xenograft studies (**Figure 3 K,L**) demonstrated increases in 5mC in NOS2 expressing xenograft tumors, which confirms that our *in vitro* and *in vivo* model systems are similar in terms of NO concentrations and downstream effects.

11. *What is the expected concentration half constant for NO to form the dinitrosyl iron complex? How does that compare with the concentration-dependent response on enzyme activity? Is there any way to ascertain the stoichiometry of the iron nitrosylation? My understanding is the iron in soluble guanylyl cyclase has very high affinity for NO and would be activated by much lower concentration of NO that is needed to change enzyme activity here. So, I am wondering if physiologic NO can indeed target the enzymes studied here, or is it simply only relevant when iNOS is expressed during inflammatory disease scenarios?*

If we understand the reviewer correctly, they are asking what the NO concentration is at which the dinitrosyl is half saturated, as in a K_d value or IC_{50} . What we know is that TET inhibition by NO results from DNIC assembly at the catalytic iron atom. As a DNIC contains 2 NO molecules, the question becomes does the DNIC form on each enzyme immediately, or do the NO molecules have differing affinities, with the second NO molecule binding more slowly or only at higher concentrations? This question was in fact our rationale for measuring DNIC formation on TET2 at two time points (1 and 20 min., **Figure 2A** and **Extended Data Figure 2**). Our hypothesis was that we would first see the NO mono-nitrosyl at 1 min and the DNIC at 20 min. However, we observed the DNIC at both time points, suggesting the second NO molecule binds with equal or greater affinity than the first, as reflected in the text. More extensive concentration and rapid trapping methods would be required to further characterize these steps, which we plan in future studies.

For these reasons, in the current manuscript we focused on elucidating the mechanistic details of a novel epigenetic signaling role for NO by demonstrating how NO inhibits DNA demethylases, and what the downstream consequences of inhibition are. Ongoing studies by our group are focused on more detailed mechanistic aspects to determine the molecular parameters that may influence how NO reacts (affinity and orientation of NO) with TET (and other Fe(II)/2-OG-dependent enzymes). Although our data suggests that NO reacts with TET first to prevent cofactor binding, we suspect that inhibition by NO will also be influenced by the type of DNA substrate (5mC vs. 5hmC vs. 5fC) as well as the cofactor concentrations.

It is true that sGC is activated by low nM concentration of NO and it is probably the most sensitive biological target for NO. In this manuscript we discuss our findings as they relate to pathological situations, most notably cancer, where NOS2 is considered to be the major isoform associated with deleterious phenotypes. We suspect

that the epigenetic effects of NO on DNA methylation will extend beyond cancer to include chronic inflammation, infection, and any other diseases where NOS2/NO is a major etiologic factor. We do not know, however, as the reviewer points out, whether NO may regulate DNA demethylation under homeostatic physiology conditions. In addition to sGC activation, NO has other purported roles to regulate protein function and cell phenotype through various alternative mechanisms (i.e., S-nitrosation). The signaling actions of NO are extremely context specific and will be dictated by the location of NO, its concentration, and its duration of exposure as well as the concentrations and types of intercellular NO targets, reactants, and scavengers. Therefore, NO most likely activates multiple concurrent pathways to varying degrees. Future directions of this project are to examine “normal” cells and animal models to determine if NO-mediated TET regulation may play a role in physiology or development.

12. The aggressive cancer phenotype induced by chronic NO exposure correlates with alteration in methylation changes at specific loci. There will be many other signaling events and changes in these cells during this treatment, and so I am wondering is anything can be done to connect these together to strengthen causality. For example, expression changes have been detected - can some of these proteins that may rationally explain the aggressive phenotype be targeted to attenuate the transition? For example, a protein or pathway that is upregulated and may explain the phenotype, might be genetically or pharmacologically targeted and it could be determined if this attenuates the aggressive cancer phenotype?

This is an important point. As our sequencing data demonstrated, hundreds of genes were transcriptionally up- or down-regulated in response to NO (**Figure 4F**). What was somewhat surprising, however, is that the overlap in common genes regulated by NO in two phenotypically distinct TNBC cell types was low (90 total, ~12%). This transcriptional plasticity suggests that different genotypic or phenotypic backgrounds are major determinants in the regulation of NO-responsive genes. Despite the low overlap in common NO-regulated genes, pathway analysis demonstrated that overall, similar tumor-permissive pathways were being activated in both cell lines. Although we have considered genetically manipulating one or more genes involved in a specific pathway/phenotype (e.g., migration), as the reviewer suggests, we suspect that deleting or overexpressing one gene may not be sufficient to attenuate a phenotype.

We should note that data we obtained from a limited CRISPR kinase knockout screen we conducted under similar conditions as our TET studies (10-day low physiologic NO-treated TNBC cells) showed that of the ~6,000 genes that were knocked out in the screen only 6 were negatively selected by NO. None of these 6 kinases were associated with 5mC or 5hmC at their promoter regions or gene regulatory loci. These findings suggest NO elicits a compound genetic phenotype and that knocking a single gene would likely prove futile in terms of attenuating NO-dependent phenotypes. Although out of the scope of this manuscript, the determination of NO-regulated epigenetic pathways that directly control tumor cell phenotype is an active area of investigation in our lab. We hope to conduct a genome wide CRISPR knockout screen using NO-treated cells to increase the probability of identifying these specific genes (that could be targeted therapeutically). In this manuscript, we stop short of calling NO-mediated changes in DNA methylation a bona fide signaling mechanism and suggest that it may instead represent a more generalized pathologic dysregulation of gene expression programs.

Reviewer #4 (Remarks to the Author):

13. The manuscript by Palczewski and coworkers describes the interaction of NO with ten-eleven translocation DNA demethylases. The data extend logically from purified samples to cultured cells to significant in vivo analysis. While the findings are impactful, the manuscript itself is sometimes too cursory to the point of being confusing. A key example is a lack of discussion of the NO donors used, including the relationship to NO concentrations. Another is the use of a neutral complex in modeling, presumably by adding a random electron. There is also a surprising dearth of references in the introduction. The figures are also quite small. In general, however, the experiments are logical, the data are informative and the conclusions are significant with regard to the impact of NO on demethylation, especially in cancer cells. Specific comments follow.

We thank this reviewer for their thorough and constructive feedback on our manuscript. We appreciate the opportunity to address the points raised to improve the clarity and quality of our work. We acknowledge that the discussion of the NO donors used in our study was initially insufficiently detailed. In the revised manuscript we have included a more comprehensive description of the NO donors utilized, including their chemical properties, mechanisms of NO release, and relevance to physiological NO concentrations (see new **Figures 1D and Extended Data Figure 1** and the descriptions in the results section as well as response #3 to reviewer 1 above).

Our modeling did indeed use a neutral complex, and we appreciate the reviewer's attention to this detail (we address this below). Moreover, in response to the reviewers concerns, we have modified the introduction for clarity including the addition of appropriate references. We have updated and resized many of the figures to ensure they are more legible (**Figure 4** has now been broken up into new **Figures 4** and **5**). **Figure 6** (previously **Figure 5**) has been completely recreated for clarity.

14. *The chemistry of NO is rather cursorily described in the introduction, including the fact that NO does not directly induce nitrosation.*

We agree that a more thorough description of the chemistry of NO is warranted in the introduction. We have added the following text:

“Despite the importance of methylated DNA bases and their oxidized derivatives in controlling normal gene expression or facilitating oncogenic transcriptional states, there is a lack of mechanistic knowledge regarding endogenous molecular regulators of DNA methyl-modifying machinery. One such potential regulator, that has not been extensively studied in the context of epigenetics, is nitric oxide (NO), an endogenously produced free radical signaling molecule known for its diverse physiological roles, including neurotransmission, immune responses, and vascular homeostasis¹². NO is a reactive molecule and has a short biological half-life¹³, yet under biological conditions its unique chemistry restricts its reactions to only metals and other free radicals, including oxygen (O₂)¹⁴. To mediate its signaling effects, NO primarily targets metalloproteins, particularly those coordinating iron at their active sites, regulating their functions through the formation of complexes such as iron nitrosyl (Fe(II)=NO) in heme proteins, or dinitrosyl iron complexes (DNIC) in non-heme proteins¹⁵. Alternatively, NO can indirectly regulate enzyme function posttranslationally through complex reactions that form adducts containing nitrogen oxide functional groups, such as S-nitrosothiols (RSNO)^{12,16,17}. Regardless of NO’s mechanism of action, the specificity of NO signaling, which can either activate or inhibit protein activity, is governed by various microenvironmental factors, including the type and abundance of target proteins, the concentration and duration of NO synthesis, the redox state of the cell, and local oxygen concentrations. These parameters influence the chemical reactions of NO which result in signaling that is both highly regulated and extremely context specific.

In addition to the critical roles NO plays in maintaining normal physiological homeostasis, dysregulated NO synthesis and signaling are implicated in various pathological conditions, particularly cancers. Elevated NO production, often driven by upregulation of the inducible form of nitric oxide synthase (NOS2), is associated with altered tumor gene expression, poor patient outcomes, increased mortality, and resistance to chemotherapy across various cancer types, including triple-negative breast cancer (TNBC)¹⁸⁻²⁸, lung²⁹⁻³¹, prostate^{32,33}, brain³⁴, colon^{35,36}, melanoma³⁷⁻³⁹, and liver^{40,41}. However, despite major advances in our understanding of NO signaling, it remains insufficient to fully explain the complexity of NO’s diverse effects in disease.

Dysregulated NO production and perturbations in DNA methylation patterns are both associated with many of the same pathologies, yet a direct causal mechanism linking NO synthesis to altered DNA methylation patterns has not been described. Our previous research demonstrated a novel role for NO as an endogenous epigenetic regulator of gene expression by controlling histone post-translational modifications^{42,43} and mRNA methylation (m⁶A) patterns⁴⁴. We demonstrated mechanistically that physiologic NO concentrations could inhibit the catalytic activities of histone lysine demethylases (KDM) and the RNA demethylase (fat mass and obesity associated protein (FTO)⁴⁵⁻⁴⁷) by forming DNIC complexes at their active sites. Since KDM and FTO demethylases belong to the same family of Fe(II)/2-oxoglutarate (OG)-dependent oxygenases as the DNA demethylases TET and ALKBH2, we hypothesized that NO would similarly inhibit TET and ALKBH2 to change DNA methylation patterns.

This study investigates NO as an endogenous regulator of DNA methylation, with a focus on triple-negative breast cancer (TNBC) models. This focus is driven by the observation that patients with this molecular subtype that harbor NOS2-expressing tumors have a significantly higher mortality rate than patients with tumors that do not express NOS2²⁷. Although the ability to correctly assign the transcriptional regulation of a particular gene to the local (or distant) presence of DNA methylation has remained surprisingly limited⁴⁸, our data suggest that deleterious phenotypic effects of NO are partially determined by transcriptional reprogramming mediated by TET inhibition and the resultant changes in the DNA methyl landscape.”

15. *Results should be removed from the introduction.*

Any reference to results have now been removed from the introduction section in the revised manuscript.

16. *In Figure 1A, inclusion of stoichiometric analysis would be beneficial.*

We appreciate the reviewer's suggestions and agree that stoichiometric analysis would be interesting for some of the experiments. NO is a reactive free radical and in aqueous solution it has a short half-life. The ability to add precise known amounts of pure NO to an enzymatic reaction becomes challenging as the NO concentration changes dynamically. However, for both TET2 and ALKBH2 we were able to measure dose-dependent inhibition by NO and were able to calculate an IC₅₀ for the NO-donor compounds used in each experiment. Although NO-donor compounds are very useful because of their predictable release kinetics, they present challenges as the steady-state NO concentration cannot be accurately extrapolated from the starting concentration of the NO-donor. When using NO-donor compounds in solution, the steady-state NO concentration is determined by the difference in the rate of NO liberation from the donor and the rate of NO consumption (predominantly autooxidation in aqueous solution). Given by:

$$\frac{d[NO]}{dt} = k_1[D]e_{NO} - k_2[O_2][NO]^2$$

Where [NO] is the concentration of NO at given time t , k_1 is the rate constant for the decomposition of the donor, [D] is the concentration of the donor at time t , e_{NO} is the factor representing moles of NO release per mole of donor, $[O_2]$ is the oxygen concentration, and k_2 is the rate constant for oxidation of NO by oxygen (autooxidation)⁴⁶. As mentioned above, (**response #3 to Reviewer 1**), we used these equations to solve a system of differential equations that simulated theoretical steady-state NO concentrations over time based on the concentration and half-life of each NO donor. This approach was used to estimate the steady-state NO concentrations for each NO-donor concentration in **Figure 1D**. Based on our simulations, the IC₅₀ for [NO]_{SS} would be approximately 3 μ M for isolated TET2 enzyme (at 5 μ M) under our assay conditions for **Figure 1A, B, C**. The IC₅₀ for ALKBH2 (under conditions of **Fig. 1H**) would be approximately 700-800 nM (**See new Extended Data Figure 1D**). But our estimates suggest that the "minimum inhibitory" NO concentration may be much lower (**Fig. 1D.3**). Importantly, these IC₅₀s for TET2 and ALKBH2 for NO represent a steady-state flux of NO, not a static NO concentration. We acknowledge that our calculations provide only general estimates as we have potentially oversimplified a complex reaction system. However, our calculated steady-state NO concentrations are in the range of what we have measured experimentally from the same NO-donor compounds^{3-5,13}. If our proposed mechanism is correct and the inhibitory mechanism of TET results from DNIC formation, it would require 2 NO molecules for every TET protein. Also, we are aware that that if this inhibition is competitive with substrates for the active site, the actual IC₅₀ for NO could be different as it could be influenced by the concentrations of the enzyme and substrate. As this calculated IC₅₀ for NO is close to half the enzyme concentration, it might indicate a specific competitive interaction under our assay conditions, but this would require a more detailed stoichiometric and kinetic analysis to confirm, which we plan to pursue in future studies.

17. *Too many significant figures are reported, beginning with Figure 1B.*

We agree and have reduced the number of significant figures.

18. *What is the error in Figure 1C?*

Error bars are now included in **Figure 1C**.

19. *Were controls performed with expired SPER/NO?*

The reviewer makes an important point, and we appreciate our oversight being pointed out. Yes, these experiments were control-conducted, and we should have mentioned these data in our manuscript. All cellular and kinetic studies were initially conducted with at least one expired NONOate control sample (an expired NONOate control sample was included at a concentration that matched the concentration of the highest NONOate concentration for each experiment). Due to cost constraints, we did not use expired NONOates for the large-scale sequencing studies. However, we know from our previous work that expired NONOates do not

inhibit or activate other Fe(II)/2-OG-dependent proteins¹³. Also, for the aforementioned CRISPR knockout study that we recently conducted (mentioned above Reviewer 3, #12) we used expired NONOates (10 day treatment) and did not measure any significant effects on gene expression or phenotype.

20. *In Figure 1D, does the negative sign indicate that DEA/NO was present for 3 hours, just expired during this time period? Again, stoichiometric analysis would be beneficial for this data set. Is there a reason that 50% inhibition is observed for both donors at 1 h? Is a longer time period for the SPER/NO experiment feasible?*

We regret that **Figure 1D** and its description were confusing. This figure has been completely redone with the inclusion of new data in hopes of clarifying the concept of reversibility and stoichiometry (Please refer to response #3 to reviewer 1 above).

21. *Starting at line 162, it would be beneficial to describe the bonding site. That being said, binding of iron by NO is well-known and thus does not have to be rationalized.*

We have added more information regarding the bonding sites:

“Based on the DFT results, further modeling was done for the DNIC core and the crystal structure of TET2 in complex with N-oxalylglycine (OGA), a 2-oxoglutarate analog⁵⁸ (PDB ID 4NM6, **Fig. 2E**). The OGA complex has iron coordinated to two histidine residues and an aspartate, a typical arrangement for Fe(II)/2-OG-dependent DNA demethylases. OGA is coordinated to the iron through two oxygens and stabilized through hydrogen bonding to Arg 1261 and a second arginine (not shown). A water molecule occupies the sixth coordination site in the structure, which is replaced by dioxygen during catalysis. In the DNIC model, the NO molecules replace OGA and occupy similar positions to the OGA-coordinating oxygens. One of these is close to Arg 1261, which may hydrogen bond to the NO and stabilize overall negative charge buildup on the NO ligands in the DNIC (**Fig. 2F**).

22. *Figure 2A: a spectrum for the enzyme alone should be included.*

New data has been added to address this and:

“Moreover, upon NO removal the TET2-DNIC EPR signal disappeared after continued incubation (>3h) (**Extended Data Figure 2B**), which coincided to gain of TET2 demethylase activity (**Fig. 1.3**).”

23. *Starting at line 184, more details need to be provided here including assignment as a triplet.*

We apologize for the lack of clarity and have added the text below:

“Among all the spin states assessed for $\text{Fe}(\text{OAc})(\text{Im})_2(\text{NO})_2(\text{OH}_2)$, the triplet (intermediate spin, 2 unpaired electrons) is predicted to be the lowest in free energy relative to the high spin (quintet) and low spin (singlet) states⁵⁶. DNIC, with an extra electron in the complex, is a typical configuration and has previously been described as having a triplet state⁵⁵. While the source of this extra electron is not yet known, many reductants are available in biological systems.”

24. *Starting at line 229, epigenetic inheritance is mentioned. Yet, NO leads to an increase in methylated product while the level is unchanged in the absence of NO. This suggests susceptible vs. non-susceptible demethylations sites. Analysis with NO production inhibited could be informative.*

This reviewer refers to our statement (line 229) “Under biological conditions, DNA methylation patterns are faithfully maintained over multiple cell generations as a form of epigenetic inheritance.” What we meant by this, is that the strict definition of an “epigenetic trait” is that it must be mitotically heritable (cell to cell and or transgenerational). We were trying to present rationale for developing a long-term NO treatment model to study epigenetic traits after multiple cell generations. However, we suspect that all DNA methylation sites are heritable

regardless of whether they are present basally or induced at new sites by NO. The reviewer makes an interesting point (susceptible vs. non-susceptible sites), in that not all DNA methylation sites are targeted equally by TET.

To address the reviewers concerns we have added the following to the discussion:

“The fact that not all CpG sites are increasing equally in response to NO suggests that there are “hot-spots” of TET activity that may reflect TET’s genomic location. We suspect that the sites of NO-dependent CpG methylation correspond to TET target regions of DNA in open chromatin where transcriptional activity is occurring or at regulatory regions that control transcriptional activity.”

25. *Figure 3E would again benefit from stoichiometric analysis or at least discussion relative to physiological levels.*

Although it is practical to calculate theoretical NO concentrations for NO-donors in solutions of isolated enzymes, it is much more challenging to make accurate predictions of NO concentrations for *in vivo* cell culture experiments. The reason being is that in cell culture there are many more NO-consumptive mechanisms present in the system that affect steady-state NO concentrations. These include the cell culture flask surface area, NO volatilization, cellular NO metabolism, cell density, O₂ gradients and O₂ concentration, reactions with media components, and metal contamination. Regardless of these factors, we try to expose our cells to physiologically relevant NO concentrations. Using chemiluminescent or electrochemical detection we have measured steady-state NO concentrations in the media of cells treated with NO-donor compounds. In general, the [NO] in cell culture is much less than what would be predicted from our modeling of NO concentrations in an ideal isolated system. For example, when 1,000 μM DETA/NO was added to cells in culture, we measured a steady state NO concentration over 24 h of 400-800 nM¹³ and 500 μM DETA/NO resulted in a steady state NO concentration of 200-500 nM⁴. For all cell culture experiments in the current manuscript, the highest DETA/NO concentration we use was 100 μM, suggesting that the cells were experiencing low nM NO concentrations. Another point in regard to stoichiometric analysis of cell culture experiments is that we do not know the intracellular concentration of TET enzymes. Despite these factors, we are confident that we are treating our cells with NO under physiologically / pathologically-relevant conditions. See answer 9 and 10 for Reviewer #3. We agree that a more detailed discussion of NO concentrations is warranted throughout the manuscript.

26. *Figure 3G would benefit from some work with ChemDraw.*

We appreciate this suggestion, **Figure 3G** has been updated with structures from ChemDraw.

27. *In the experiment starting on line 296, NO levels should be described.*

See response #25 above.

28. In the discussion of previous studies, are there any clear differences, such as NO concentration, tissue type, etc?

Yes, there are actually some distinct differences between these studies which could explain some differences in the results. In the first study (Switzer et al.), although both that study and ours used the same cell lines, they used a much higher concentration of DETA/NO (300 μM), whereas we used concentrations of 100 μM and lower. The use of higher NO concentrations may explain why they observed more nitrosative chemistry. The second study was conducted on mouse aortas, not on cancerous tissue. The final study we referenced involved different cell types, specifically the gastric cancer cell lines HSC41 and TMK1 and also used different NO donors (NOC18 or SNAP). SNAP is transnitrosating agent and not a true NO donor, which could have dramatically different effects on cell signaling.

29. Line 390 – with NO bound perhaps would be better.

We have changed the wording.

30. In response to the final discussion, it would be informative, although not necessary to this study, to compare to nontumorigenic cell lines such as MCF-10.

We agree, and this is a focus of future studies in our labs. In general, we suspect that NO would have similar effects on TET inhibition and altered DNA methylation in most cell types (cancer and normal). Factors that may influence NO-dependent methylation in other cell types would be the expression levels of the TET enzymes, the metabolic activity of the cells, growth rate, genetic background, cofactor availability, and overall transcriptional activity.

References

1. Belle R, Sarac H, Salah E, et al. Focused Screening Identifies Different Sensitivities of Human TET Oxygenases to the Oncometabolite 2-Hydroxyglutarate. *J Med Chem.* 2024;67(6):4525-4540.
2. Putiri EL, Tiedemann RL, Thompson JJ, et al. Distinct and overlapping control of 5-methylcytosine and 5-hydroxymethylcytosine by the TET proteins in human cancer cells. *Genome Biol.* 2014;15(6):R81.
3. Thomas DD, Miranda KM, Espey MG, et al. Guide for the use of nitric oxide (NO) donors as probes of the chemistry of NO and related redox species in biological systems. *Methods Enzymol.* 2002;359:84-105.
4. Hickok JR, Sahni S, Shen H, et al. Dinitrosyliron complexes are the most abundant nitric oxide-derived cellular adduct: biological parameters of assembly and disappearance. *Free Radic Biol Med.* 2011;51(8):1558-1566.
5. Hickok JR, Sahni S, Mikhed Y, Bonini MG, Thomas DD. Nitric oxide suppresses tumor cell migration through N-Myc downstream-regulated gene-1 (NDRG1) expression: role of chelatable iron. *J Biol Chem.* 2011;286(48):41413-41424.
6. Mariani CJ, Vasanthakumar A, Madzo J, et al. TET1-mediated hydroxymethylation facilitates hypoxic gene induction in neuroblastoma. *Cell Rep.* 2014;7(5):1343-1352.
7. He P, Lei J, Zou LX, et al. Effects of hypoxia on DNA hydroxymethylase Tet methylcytosine dioxygenase 2 in a KG-1 human acute myeloid leukemia cell line and its mechanism. *Oncol Lett.* 2021;22(4):692.
8. Prasad P, Mittal SA, Chongtham J, Mohanty S, Srivastava T. Hypoxia-Mediated Epigenetic Regulation of Stemness in Brain Tumor Cells. *Stem Cells.* 2017;35(6):1468-1478.
9. Lin G, Sun W, Yang Z, Guo J, Liu H, Liang J. Hypoxia induces the expression of TET enzymes in HepG2 cells. *Oncol Lett.* 2017;14(6):6457-6462.
10. Wu MZ, Chen SF, Nieh S, et al. Hypoxia Drives Breast Tumor Malignancy through a TET-TNFalpha-p38-MAPK Signaling Axis. *Cancer Res.* 2015;75(18):3912-3924.
11. Cao JZ, Liu H, Wickrema A, Godley LA. HIF-1 directly induces TET3 expression to enhance 5-hmC density and induce erythroid gene expression in hypoxia. *Blood Adv.* 2020;4(13):3053-3062.
12. Hickok JR, Vasudevan D, Jablonski K, Thomas DD. Oxygen dependence of nitric oxide-mediated signaling. *Redox Biol.* 2013;1:203-209.
13. Thomas DD, Espey MG, Ridnour LA, et al. Hypoxic inducible factor 1alpha, extracellular signal-regulated kinase, and p53 are regulated by distinct threshold concentrations of nitric oxide. *Proc Natl Acad Sci U S A.* 2004;101(24):8894-8899.
14. Thomas DD, Ridnour LA, Espey MG, et al. Superoxide fluxes limit nitric oxide-induced signaling. *J Biol Chem.* 2006;281(36):25984-25993.
15. Sandau KB, Fandrey J, Brune B. Accumulation of HIF-1alpha under the influence of nitric oxide. *Blood.* 2001;97(4):1009-1015.
16. McCarty G, Loeb DM. Hypoxia-sensitive epigenetic regulation of an antisense-oriented lncRNA controls WT1 expression in myeloid leukemia cells. *PLoS One.* 2015;10(3):e0119837.

17. Koh KP, Yabuuchi A, Rao S, et al. Tet1 and Tet2 regulate 5-hydroxymethylcytosine production and cell lineage specification in mouse embryonic stem cells. *Cell Stem Cell*. 2011;8(2):200-213.
18. Burr S, Caldwell A, Chong M, et al. Oxygen gradients can determine epigenetic asymmetry and cellular differentiation via differential regulation of Tet activity in embryonic stem cells. *Nucleic Acids Res*. 2018;46(3):1210-1226.
19. Hickok JR, Vasudevan D, Antholine WE, Thomas DD. Nitric oxide modifies global histone methylation by inhibiting Jumonji C domain-containing demethylases. *J Biol Chem*. 2013;288(22):16004-16015.
20. Palczewski MB, Kuschman HP, Bovee R, Hickok JR, Thomas DD. Vorinostat exhibits anticancer effects in triple-negative breast cancer cells by preventing nitric oxide-driven histone deacetylation. *Biol Chem*. 2021;402(4):501-512.
21. Vasudevan D, Hickok JR, Bovee RC, et al. Nitric Oxide Regulates Gene Expression in Cancers by Controlling Histone Posttranslational Modifications. *Cancer Res*. 2015;75(24):5299-5308.
22. Hickok JR, Thomas DD. Nitric oxide and cancer therapy: the emperor has NO clothes. *Curr Pharm Des*. 2010;16(4):381-391.
23. Vasudevan D, Thomas DD. Insights into the diverse effects of nitric oxide on tumor biology. *Vitamins and hormones*. 2014;96:265-298.
24. Loibl S, Buck A, Strank C, et al. The role of early expression of inducible nitric oxide synthase in human breast cancer. *European journal of cancer*. 2005;41(2):265-271.
25. De Paepe B, Verstraeten VM, De Potter CR, Bullock GR. Increased angiotensin II type-2 receptor density in hyperplasia, DCIS and invasive carcinoma of the breast is paralleled with increased iNOS expression. *Histochemistry and cell biology*. 2002;117(1):13-19.
26. Heinecke JL, Ridnour LA, Cheng RY, et al. Tumor microenvironment-based feed-forward regulation of NOS2 in breast cancer progression. *Proc Natl Acad Sci U S A*. 2014;111(17):6323-6328.
27. Switzer CH, Cheng RY, Ridnour LA, Glynn SA, Ambs S, Wink DA. Ets-1 is a transcriptional mediator of oncogenic nitric oxide signaling in estrogen receptor-negative breast cancer. *Breast cancer research : BCR*. 2012;14(5):R125.
28. Ridnour LA, Barasch KM, Windhausen AN, et al. Nitric oxide synthase and breast cancer: role of TIMP-1 in NO-mediated Akt activation. *PloS one*. 2012;7(9):e44081.
29. Switzer CH, Ridnour LA, Cheng R, et al. S-Nitrosation Mediates Multiple Pathways That Lead to Tumor Progression in Estrogen Receptor-Negative Breast Cancer. *Forum on immunopathological diseases and therapeutics*. 2012;3(2):117-124.
30. Switzer CH, Glynn SA, Ridnour LA, et al. Nitric oxide and protein phosphatase 2A provide novel therapeutic opportunities in ER-negative breast cancer. *Trends in pharmacological sciences*. 2011;32(11):644-651.
31. Glynn SA, Boersma BJ, Dorsey TH, et al. Increased NOS2 predicts poor survival in estrogen receptor-negative breast cancer patients. *J Clin Invest*. 2010;120(11):3843-3854.
32. Prueitt RL, Boersma BJ, Howe TM, et al. Inflammation and IGF-I activate the Akt pathway in breast cancer. *International journal of cancer. Journal international du cancer*. 2007;120(4):796-805.
33. Liu PF, Zhao DH, Qi Y, et al. The clinical value of exhaled nitric oxide in patients with lung cancer. *Clin Respir J*. 2018;12(1):23-30.
34. Zhang L, Liu J, Wang X, et al. Upregulation of cytoskeleton protein and extracellular matrix protein induced by stromal-derived nitric oxide promotes lung cancer invasion and metastasis. *Current molecular medicine*. 2014;14(6):762-771.
35. Gao X, Xuan Y, Benner A, Anusruti A, Brenner H, Schottker B. Nitric Oxide Metabolites and Lung Cancer Incidence: A Matched Case-Control Study Nested in the ESTHER Cohort. *Oxid Med Cell Longev*. 2019;2019:6470950.

36. Lee KM, Kang D, Park SK, et al. Nitric oxide synthase gene polymorphisms and prostate cancer risk. *Carcinogenesis*. 2009;30(4):621-625.
37. Erlandsson A, Carlsson J, Andersson SO, et al. High inducible nitric oxide synthase in prostate tumor epithelium is associated with lethal prostate cancer. *Scand J Urol*. 2018;52(2):129-133.
38. Fahey JM, Korytowski W, Girotti AW. Upstream signaling events leading to elevated production of pro-survival nitric oxide in photodynamically-challenged glioblastoma cells. *Free Radic Biol Med*. 2019;137:37-45.
39. Puglisi MA, Cenciarelli C, Tesori V, et al. High nitric oxide production, secondary to inducible nitric oxide synthase expression, is essential for regulation of the tumour-initiating properties of colon cancer stem cells. *J Pathol*. 2015;236(4):479-490.
40. de Oliveira GA, Cheng RYS, Ridnour LA, et al. Inducible Nitric Oxide Synthase in the Carcinogenesis of Gastrointestinal Cancers. *Antioxidants & redox signaling*. 2017;26(18):1059-1077.
41. Goncalves DA, Xisto R, Goncalves JD, et al. Imbalance between nitric oxide and superoxide anion induced by uncoupled nitric oxide synthase contributes to human melanoma development. *Int J Biochem Cell Biol*. 2019;115:105592.
42. Massi D, De Nisi MC, Franchi A, et al. Inducible nitric oxide synthase expression in melanoma: implications in lymphangiogenesis. *Mod Pathol*. 2009;22(1):21-30.
43. Lopez-Rivera E, Jayaraman P, Parikh F, et al. Inducible nitric oxide synthase drives mTOR pathway activation and proliferation of human melanoma by reversible nitrosylation of TSC2. *Cancer research*. 2014;74(4):1067-1078.
44. Wang R, Li Y, Tsung A, et al. iNOS promotes CD24(+)CD133(+) liver cancer stem cell phenotype through a TACE/ADAM17-dependent Notch signaling pathway. *Proc Natl Acad Sci U S A*. 2018;115(43):E10127-E10136.
45. Park YH, Shin HJ, Kim SU, et al. iNOS promotes HBx-induced hepatocellular carcinoma via upregulation of JNK activation. *Biochem Biophys Res Commun*. 2013;435(2):244-249.
46. He W, Frost MC. Direct measurement of actual levels of nitric oxide (NO) in cell culture conditions using soluble NO donors. *Redox Biol*. 2016;9:1-14.
47. Pereira JC, Iretskii AV, Han RM, Ford PC. Dinitrosyl iron complexes with cysteine. Kinetics studies of the formation and reactions of DNICs in aqueous solution. *J Am Chem Soc*. 2015;137(1):328-336.
48. Speelman AL, Zhang B, Silakov A, et al. Unusual Synthetic Pathway for an Fe(NO)₂(9) Dinitrosyl Iron Complex (DNIC) and Insight into DNIC Electronic Structure via Nuclear Resonance Vibrational Spectroscopy. *Inorg Chem*. 2016;55(11):5485-5501.

RESPONSES TO THE REVIEWERS' COMMENTS

Summary

We thank reviewers 2-4 for their overwhelming support to publish the manuscript in its current state. The remaining concerns with our manuscript are from Reviewer #1, who questions our methodology and whether 5hmC can increase in cells where TET is inhibited. We provide additional evidence that this in fact does occur despite being paradoxical at first glance. In the revised manuscript, we:

- Emphasize three methods showing NO-dependent increases in 5hmC levels, ELISA, dot blot and sequencing. Additionally, we provide new data below demonstrating that 5hmC increases in DNA from NO treated cells as measured by LC-MS/MS.
- Highlight other publications with similar results.
- Emphasize in the discussion that some studies examining TET inhibition in cells find only increases in 5mC levels and not 5hmC, unlike the case here.
- Discuss possible mechanisms for how 5hmC increases may result from the multiple enzymes involved (TET1-3) and their independent susceptibilities to NO inhibition and substrate and intermediate affinities.
- Discuss how NO inhibition mimics hypoxia.
- Discuss the challenges with using dissolved NO gas, which persists in biological samples for only a few seconds, in cell-based experiments. Delivery of NO by NO releasing compounds or through introduction of NO synthase into cells, overcomes this challenge.

We thank reviewer #1 for his or her efforts on our behalf and hope these additional changes clarify our findings and alleviate the reviewer's concerns.

Reviewer #1 (Remarks to the Author):

1. *The authors provided a revised version of the manuscript, however they failed to address the main concern that, according to their data, inhibition of TET by NO leads to increased levels of hmC in cells. TET inhibition should reduce the levels of hmC in genomic DNA, which calls the paper's conclusions into question. One possible explanation for this puzzling result is the limitations of the methodology used in this paper. They claim that they used a "rigorous approach" to quantify hmC, however, ELISA can hardly be considered a quantitative method for hmC, and no independent confirmation of this results was provided. A golden standard in the field is isotope dilution HPLC-MS/MS to provide absolute accurate and precise levels of hmC and other epigenetic marks in cells, but this was not done in this study.*

We thank the reviewer for bringing up this issue again and we regret that our responses in the first resubmission did not provide adequate explanation. Respectfully, we would like to again highlight a few manuscripts where increases in 5hmC have been reported when TET enzymes were inhibited chemically or inhibited due to decreased substrate availability (hypoxia, O₂) under various *in vitro* and *in vivo* conditions (see references¹⁻⁸). We agree that ELISA is not a quantitative assay, which is why we reported our data in relative terms. We also agree that mass spectrometry has some advantages as it is both highly specific and also quantitative. We thank the reviewer for the reference on the use of MS for 5hmC (Guan et al, Blood Cancer Discov. 2021 Mar;2(2):146-161), which, although an impressive paper, appears to exclusively use antibody-based methods (dot blot and ELISA) for identifying 5hmC. We suspect the reviewer may have intended to recommend a different paper for MS methodology. Regardless, when we first started our studies on TET several years ago, we made sure to measure DNA cytosine adducts in cells treated with NO by mass spectrometry (method modified from⁹). These studies were conducted in collaboration with two independent laboratories where we sent DNA samples (blinded) from NO treated cells for MS analysis for modified DNA cytosine adducts. As can be seen in the figure below the results from both labs demonstrated a quantitative increase in 5hmC in the NO-treated cells compared to the cells not treated with NO. At the time we were interested in neuroblastoma which is why one of the cell types was SH-SY5Y.

Mass Spectrometry analysis of 5hmC in DNA from NO treated cells. Absolute amounts of 5hmC were measured by a quantitative LC-MS/MS assay and are displayed as 5hmC / total dC %. Cells were treated with NO gas using the NO-donor DETA/NO (500 μ M) and the DNA was isolated from the cells at the indicated time points. The DNA hydrolysates were spiked with 5-hydroxymethyl-d₂-2'-deoxycytidine-6-d₁ (896 fmol, internal standard for mass spectrometry). dC was quantified by HPLC-UV using calibration curves obtained by analyzing authentic dC standards. HPLC fractions corresponding to dC and hmdC, were analyzed by isotope dilution HPLC-ESI-MS/MS. Data from "A" and "B" were obtained from two separate laboratories. (A) MDA-MB-231 human triple negative breast cancer cells treated with NO for 12 hours. (B) SH-SY5Y human neuroblastoma cells treated with NO for 24 or 48 h. All data are averaged experimental triplicates (mean \pm S.D, n = 3). *P*-values were determined by unpaired two-tailed Student's *t*-tests.

Having definitively established that NO increased 5hmC in DNA from NO-treated cells we ultimately settled on using the ELISA method for subsequent experiments. Although not quantitative, the ELISA had advantages for our lab for being more efficient, economical, and higher throughput. We felt confident using this method as antibody-based detection methods are well-established and commonly used by many labs studying DNA methylcytosine adducts (Dot blot: *Nature Communications*¹⁰⁻¹² and ELISA: *Nature Communications*¹³, *Nature Immunology*¹⁴, *Nature Metabolism*¹⁵).

To summarize our findings and methodology for 5hmC in NO treated cancer cells:

- We used 4 separate methods (two antibody-based (ELISA and dot blot), RRBS sequencing, and mass spectrometry)
- We conducted separate assays in four independent laboratories
- We measured 5hmC in multiple diverse cell types, (SH-SY5Y, MDA-MB-231, MDA-MB-468, PC-3, U251)
- We measured 5hmC at multiple time points in cells (12h – 10 days).

In all instances, independent of the method, laboratory, time point, or cell type, NO increased 5hmC in DNA from cells (relative or quantitative). We would also like to point out that the enzyme kinetic measurements in **figures 1A-C** were measured by MALDI-TOF-MS-based assay¹⁶.

We agree that it is important to acknowledge how our results differ from what other labs have reported so we added the following paragraph to the discussion:

"In cells, most studies by other groups have shown that small molecule chemical inhibitors of TET enzymes cause global increases in 5mC and decreases in 5hmC^{17,18}. However, other laboratories have reported that TET inhibitors¹, TET knock down², or decreased substrate availability (hypoxia, O₂)^{3,4} can lead to an increase in cellular 5hmC. Our cellular studies demonstrated that NO caused global, and loci-specific, increases in both 5mC and 5hmC in DNA. It is intuitive that 5mC would increase consequent to TET inhibition, yet mechanisms to explain why 5hmC also increases are less obvious. It is possible that of the 3 TET isoforms that are expressed in the cells we tested, one of the isoforms is more sensitive to inhibition by NO, but one of the other isoforms has a greater affinity for 5mC than 5hmC"

2. *The authors acknowledge that their “long-term” NO exposure experiments using the O donor DETA/NO repeatedly added it to the cells every 48h led to wide fluctuations in NO levels and could result in artifactual results.*

To clarify this point, we do not know whether steady state NO concentrations fluctuated in our studies, we simply mentioned this as one of many possible explanations for increases in 5hmC and based on our experience we suspect that this explanation is unlikely. We are one of the only labs in the world who has measured (quantified) real-time steady state NO concentrations in cells exposed to exogenous and endogenous NO sources¹⁹⁻²¹. Our lab has a reputation for our ability to quantify NO as well as treat cells with physiologically relevant NO concentrations.

3. *It is not clear why NO gas could not be used directly for these experiments.*

If we understand the question correctly, the reviewer is asking why we did not treat the cells with NO gas from a compressed NO gas tank. To our knowledge, this method is not used to deliver NO gas to cells in culture. For very specific experiments, in the past, we have used bolus injections of deaerated NO-gas saturated buffer solutions to treat cells, but this is not feasible for continuous treatment (> 5 min). At higher concentrations (μM), NO reacts rapidly by a second-order reaction with oxygen to form nitrite, which greatly decreases the half-life of NO (on the order of seconds). Therefore, to study the signaling effects of NO we exogenously deliver NO gas using NO-donor compounds (diazoniumdiolates; NONOates) which allows precise control over the amount and duration of NO exposure. These compounds deliver authentic NO gas, as opposed to some other compounds that transnitrosate without releasing any NO molecules. Our lab has meticulously characterized the NO release kinetics of diazoniumdiolates *in vitro* and in cell culture models so that we can accurately deliver physiologic/pathologic NO concentrations^{19,20,22-24}. Since bioavailable NO concentration cannot be linearly extrapolated from the concentration of the NO-donor used, we have directly measured the steady state NO concentration to verify that all our experiments are conducted at physiologic and pathologic NO concentrations ($\sim[\text{NO}]_{\text{ss}} = 10 - 200 \text{ nM}$). Additionally, in the manuscript we presented data from cells expressing nitric oxide synthase. We demonstrated that these cells both synthesize NO gas and that they also replicate the effects achieved from NO-donor compounds.

4. *The argument about hypoxia is hardly relevant here as the cells were not grown under hypoxic conditions.*

In addition to the examples from the literature we provided previously demonstrating increases in 5hmC under hypoxic conditions we should have offered a better rationale for why these studies are potentially relevant to our findings. As the reviewer correctly pointed out, our studies were not conducted under hypoxic conditions. There are, however, some similarities between NO exposure and hypoxic conditions. Hypoxia exerts biochemical and signaling effects by limiting oxygen as a substrate for numerous oxygen dependent enzymes (TET being one of many). Similarly, some of the effects of NO can be attributed to its ability to bind and replace oxygen at the active site of many enzymes that use oxygen as a substrate (such as TET). This is analogous to the common practice of using cobalt chloride (CoCl_2) as a “hypoxic mimetic”, which as we are well aware of, is inaccurate in a technical sense. Nevertheless, (HIF) prolyl hydroxylase (PH) and TET enzymes are structurally similar as they are in the same family of Fe(II)/2-OG-dependent dioxygenases and both enzymes are inhibited by NO and by hypoxia. TET is inhibited by hypoxia or NO, and HIF/PH is inhibited by hypoxia or NO.

We hope this clarifies why we mentioned hypoxia and that this puts what we stated previously in better context: “Therefore, adding NO is like taking away O_2 (hypoxia) (reviewed by us²⁵). For this reason, NO is sometimes referred to as a “hypoxia mimetic”. Although the term is not accurate in a technical sense, NO can bind to and inhibit the hypoxia-inducible factor (HIF) prolyl hydroxylase (PH) which results in HIF1 α accumulation and the induction of down-stream hypoxic responses. We^{22,23}, and others²⁶, have demonstrated that under normoxic conditions (21% O_2) NO is a potent inhibitor of PH leading to HIF-1 α accumulation. McCarthy et. al. (***PLoS One*** 2015²⁷) treated AML cell lines with cobalt chloride (CoCl_2), which is another chemical “hypoxia mimetic” that displaces non-heme iron in Fe(II)/2-OG-dependent dioxygenase enzymes, leading to loss of activity in a manner similar to NO inhibition. They reported that at 21% O_2 , CoCl_2 significantly increased 5hmC. One explanation, cited by many of the papers for increased 5hmC under hypoxia, was that HIF-1 α increases the expression levels of some TET isoforms. However, we showed no change in the expression of TET1,2,3 in response to NO, which is consistent with other studies demonstrating that expression profiles of the TETs do not in fact mirror the dynamic changes in global 5hmC levels^{28,29}.”

Reviewer #2 (Remarks to the Author):

1. *The reviewers have adequately addressed all my concerns and the manuscript is now strong.*

Reviewer #3 (Remarks to the Author):

1. *The authors have sufficiently addressed my comments with replies to my questions that were clear and rational.*

Reviewer #4 (Remarks to the Author):

1. *The authors responded in significant detail, including with inclusion of additional data, to the many points of the reviewers. As such, the manuscript has substantially improved in terms of soundness, experimental details and legibility, clarity to the reader, demonstration of significance, etc. As such, the recommendation is to publish at this point.*

We thank reviewers 2, 3, and 4 for their thorough critiques acknowledging the numerous improvements we have made to the manuscript and also for their recognition that the manuscript is strong and fit for publication.

References

1. Belle R, Sarac H, Salah E, et al. Focused Screening Identifies Different Sensitivities of Human TET Oxygenases to the Oncometabolite 2-Hydroxyglutarate. *J Med Chem*. 2024;67(6):4525-4540.
2. Putiri EL, Tiedemann RL, Thompson JJ, et al. Distinct and overlapping control of 5-methylcytosine and 5-hydroxymethylcytosine by the TET proteins in human cancer cells. *Genome Biol*. 2014;15(6):R81.
3. He P, Lei J, Zou LX, et al. Effects of hypoxia on DNA hydroxymethylase Tet methylcytosine dioxygenase 2 in a KG-1 human acute myeloid leukemia cell line and its mechanism. *Oncol Lett*. 2021;22(4):692.
4. Mariani CJ, Vasanthakumar A, Madzo J, et al. TET1-mediated hydroxymethylation facilitates hypoxic gene induction in neuroblastoma. *Cell Rep*. 2014;7(5):1343-1352.
5. Prasad P, Mittal SA, Chongtham J, Mohanty S, Srivastava T. Hypoxia-Mediated Epigenetic Regulation of Stemness in Brain Tumor Cells. *Stem Cells*. 2017;35(6):1468-1478.
6. Lin G, Sun W, Yang Z, Guo J, Liu H, Liang J. Hypoxia induces the expression of TET enzymes in HepG2 cells. *Oncol Lett*. 2017;14(6):6457-6462.
7. Wu MZ, Chen SF, Nieh S, et al. Hypoxia Drives Breast Tumor Malignancy through a TET-TNFalpha-p38-MAPK Signaling Axis. *Cancer Res*. 2015;75(18):3912-3924.
8. Cao JZ, Liu H, Wickrema A, Godley LA. HIF-1 directly induces TET3 expression to enhance 5-hmC density and induce erythroid gene expression in hypoxia. *Blood Adv*. 2020;4(13):3053-3062.
9. You C, Ji D, Dai X, Wang Y. Effects of Tet-mediated oxidation products of 5-methylcytosine on DNA transcription in vitro and in mammalian cells. *Sci Rep*. 2014;4:7052.
10. Zeng Q, Song J, Sun X, et al. A negative feedback loop between TET2 and leptin in adipocyte regulates body weight. *Nat Commun*. 2024;15(1):2825.
11. Thaler R, Khani F, Sturmlechner I, et al. Vitamin C epigenetically controls osteogenesis and bone mineralization. *Nat Commun*. 2022;13(1):5883.
12. Zhao Z, Li L, Zeng R, et al. 5mC modification orchestrates choriogenesis and fertilization by preventing prolonged ftz-f1 expression. *Nat Commun*. 2023;14(1):8234.
13. Okuda K, Nakahara K, Ito A, et al. Pivotal role for S-nitrosylation of DNA methyltransferase 3B in epigenetic regulation of tumorigenesis. *Nat Commun*. 2023;14(1):621.
14. Zhou B, Magana L, Hong Z, et al. The angiocrine Rspodin3 instructs interstitial macrophage transition via metabolic-epigenetic reprogramming and resolves inflammatory injury. *Nat Immunol*. 2020;21(11):1430-1443.

15. Merlin J, Ivanov S, Dumont A, et al. Non-canonical glutamine transamination sustains efferocytosis by coupling redox buffering to oxidative phosphorylation. *Nat Metab.* 2021;3(10):1313-1326.
16. Sudhamalla B, Dey D, Breski M, Islam K. A rapid mass spectrometric method for the measurement of catalytic activity of ten-eleven translocation enzymes. *Anal Biochem.* 2017;534:28-35.
17. Guan Y, Tiwari AD, Phillips JG, et al. A Therapeutic Strategy for Preferential Targeting of TET2 Mutant and TET-dioxygenase Deficient Cells in Myeloid Neoplasms. *Blood Cancer Discov.* 2021;2(2):146-161.
18. Dusadeemeelap C, Rojasawasthien T, Matsubara T, Kokabu S, Addison WN. Inhibition of TET-mediated DNA demethylation suppresses osteoblast differentiation. *FASEB J.* 2022;36(2):e22153.
19. Thomas DD, Miranda KM, Espey MG, et al. Guide for the use of nitric oxide (NO) donors as probes of the chemistry of NO and related redox species in biological systems. *Methods Enzymol.* 2002;359:84-105.
20. Hickok JR, Sahni S, Shen H, et al. Dinitrosyliron complexes are the most abundant nitric oxide-derived cellular adduct: biological parameters of assembly and disappearance. *Free Radic Biol Med.* 2011;51(8):1558-1566.
21. Hickok JR, Sahni S, Mikhed Y, Bonini MG, Thomas DD. Nitric oxide suppresses tumor cell migration through N-Myc downstream-regulated gene-1 (NDRG1) expression: role of chelatable iron. *J Biol Chem.* 2011;286(48):41413-41424.
22. Thomas DD, Espey MG, Ridnour LA, et al. Hypoxic inducible factor 1alpha, extracellular signal-regulated kinase, and p53 are regulated by distinct threshold concentrations of nitric oxide. *Proceedings of the National Academy of Sciences of the United States of America.* 2004;101(24):8894-8899.
23. Thomas DD, Ridnour LA, Espey MG, et al. Superoxide fluxes limit nitric oxide-induced signaling. *J Biol Chem.* 2006;281(36):25984-25993.
24. Hickok JR, Vasudevan D, Thatcher GR, Thomas DD. Is S-nitrosocysteine a true surrogate for nitric oxide? *Antioxid Redox Signal.* 2012;17(7):962-968.
25. Hickok JR, Vasudevan D, Jablonski K, Thomas DD. Oxygen dependence of nitric oxide-mediated signaling. *Redox Biol.* 2013;1:203-209.
26. Sandau KB, Fandrey J, Brune B. Accumulation of HIF-1alpha under the influence of nitric oxide. *Blood.* 2001;97(4):1009-1015.
27. McCarty G, Loeb DM. Hypoxia-sensitive epigenetic regulation of an antisense-oriented lncRNA controls WT1 expression in myeloid leukemia cells. *PLoS One.* 2015;10(3):e0119837.
28. Koh KP, Yabuuchi A, Rao S, et al. Tet1 and Tet2 regulate 5-hydroxymethylcytosine production and cell lineage specification in mouse embryonic stem cells. *Cell Stem Cell.* 2011;8(2):200-213.
29. Burr S, Caldwell A, Chong M, et al. Oxygen gradients can determine epigenetic asymmetry and cellular differentiation via differential regulation of Tet activity in embryonic stem cells. *Nucleic Acids Res.* 2018;46(3):1210-1226.

RESPONSES TO THE REVIEWERS' COMMENTS

Summary

The remaining concern from the previous submission was from Reviewer #1, who questions our methodology and whether 5hmC can increase in cells where TET is inhibited. We provided additional LC-MS data demonstrating that this in fact does occur despite being paradoxical at first glance.

Reviewer #1 (Remarks to the Author):

- 1. The authors have addressed my previous concerns by providing LC-MS data for hmC in cells treated with nitric oxide. The revised manuscript is suitable for publication.*

We thank all four reviewers for their thorough critiques acknowledging the numerous improvements we have made to the manuscript and also for their recognition that the manuscript is strong and fit for publication.